# Improving Reasoning Performance in Large Language Models via Representation Engineering

**Bertram Højer,**[*] **Oliver Jarvis,**[*] **Stefan Heinrich**
Department of Computer Science, IT University of Copenhagen, Denmark
{berh, ojar, stehe}@itu.dk

## Abstract

Recent advancements in large language models (LLMs) have resulted in increasingly anthropomorphic language concerning the ability of LLMs to reason. Whether *reasoning* in LLMs should be understood to be inherently different is, however, widely debated. We propose utilizing a representation engineering approach wherein model activations are read from the residual stream of an LLM when processing a reasoning task. The activations are used to derive a control vector that is applied to the model as an inference-time intervention, modulating the representational space of the model, to improve performance on the specified task. We publish the code for deriving control vectors and analyzing model representations.[1] The method allows us to improve performance on reasoning benchmarks and assess how control vectors influence the final logit distribution of a model via metrics such as KL divergence and entropy. We apply control vectors to Mistral-7B-Instruct and a range of Pythia models on an inductive, a deductive and mathematical reasoning task. We show that an LLM can, to a certain degree, be controlled to improve its perceived reasoning ability by modulating activations. The intervention is dependent upon the ability to reliably extract the model's typical state when correctly solving a task. Our results suggest that reasoning performance can be modulated in the same manner as other information-processing tasks performed by LLMs and demonstrate that we are capable of improving performance on specific tasks via a simple intervention on the residual stream with no additional training.

## 1 Introduction

Many recent developments in the study of artificial intelligence and more specifically large language models (LLMs) have focused on improving their ability to solve *reasoning* tasks. The notion of *reasoning* is, however, notoriously hard to ground and define (Chollet, 2019; Pavlick, 2023). While various forms of reasoning such as inductive, deductive and abductive reasoning are well-defined, they typically refer to high-level processes – often associated with System 2 thinking – as opposed to fundamental computational mechanisms (Johnson-Laird, 2008). The increasing scale of LLMs and better data-curation have yielded better results on standard benchmark datasets leading to research into the reasoning strategies employed by LLMs; this research generally focuses on model outputs as opposed to internal states (e.g. Mondorf & Plank, 2024). We thus cannot conclude anything with regards to the processes or representational learning dynamics related to the ability of an LLM to solve reasoning tasks. We propose modeling the typical representations of a simple "reasoning process" and using those representations to improve reasoning performance on three tasks.

Recent work has shown that it is possible to induce specific types of behavior by manually modulating the internal state of a model. Researchers have assessed editing LLM knowledge in MLP layers of transformers (Meng et al., 2023a;b) and have mapped entire computational circuits

---

[*]Equal contributions.
[1]code: https://github.com/bertramhojer/improve-reasoning-iclr-2025

within models (Wang et al., 2022). More interestingly, it has been shown that we do not need to investigate specific MLP or attention layers, but can instead look at the residual stream of an LLM. By modulating the residual stream in a meaningful way it is possible to induce different types of "behavioral" traits such as honesty, truthfulness and emotional valence. These types of behavior can be conceived as *directions* in the representational space of an LLM (Liu et al., 2023; Hendel et al., 2023; Todd et al., 2024). Additionally, modulating the residual stream has also been used to improve the ability of transformer models trained to play the board games chess and othello (Karvonen, 2024; Nanda et al., 2023). A similar approach was utilized by Anthropic for their *Golden Gate Claude* model in which they steered an LLM towards "acting" as the Golden Gate bridge (Templeton et al., 2024). The above research highlights the efficacy of the approach as well as the broad potential applications.

As a novel contribution, we directly analyze the ability to improve "reasoning" using a representation engineering approach. We specifically assess whether the residual stream of LLMs contains valuable (and actionable) information as to a model's reasoning ability and whether it can be used to improve it, as has been done for other types of model "behavior". Concretely, we extract activations from the hidden dimension of LLMs. We then create a *control vector* on the basis of these activations and assess whether our inference-time intervention can improve reasoning ability on and across the specific set of tasks we analyze.

The main contribution of this paper is thus an analysis of the effectiveness of perturbing the residual stream in order to improve a model's performance on a set of reasoning-related tasks. We particularly address whether the method allows us to find a direction in the representational space of LLMs related to *reasoning*, and how the intervention affects the representational space of a model.

## 2 APPROACH

We employ a representation engineering approach, deriving control vectors based on typical model representations when performing the specified reasoning task. This allows us to assess whether one can induce better reasoning performance similarly to how one can adjust e.g. the emotional valence of model outputs (Zou et al., 2023).

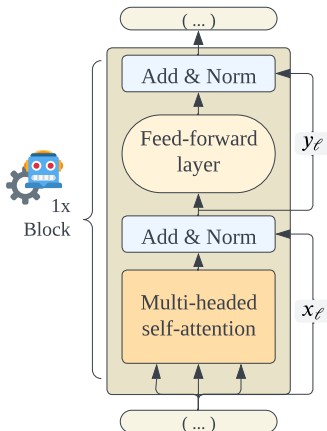
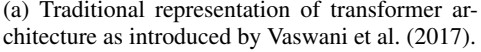
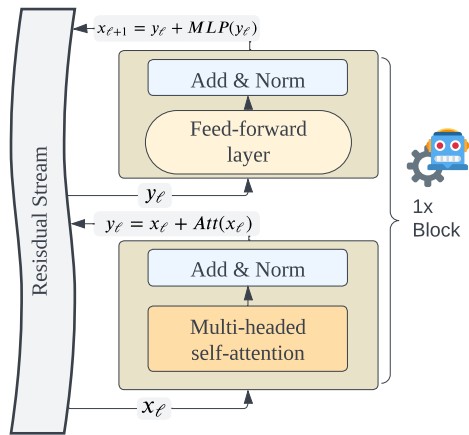

(a) Traditional representation of transformer architecture as introduced by Vaswani et al. (2017).

(b) Conceptual reframing of the transformer architecture as discussed by Elhage et al. (2021).

Figure 1: A conceptual reframing of transformer information processing emphasizing the importance of the residual stream. As opposed to focusing on the operations done by each computational block, this framing emphasizes each computational block as reading from and writing to a *residual stream* that is represented in the hidden dimension of a transformer model.

In most LLMs, the transformer architecture comprises an embedding layer followed by $n$ computational blocks finally followed by an output layer of token logits from which a new token is generated

via a softmax function over the logits (Vaswani et al., 2017). Following arguments by Elhage et al. (2021) as well as recent related research (Templeton et al., 2024), we highlight the residual stream as a key *component* of the transformer, and emphasize that the fundamental operation of the transformer is representation transformation. This conceptual reframing is illustrated in Figure 1. The operations done by a single layer in any given transformer can be described as in Equation 1 without the term $\mathbf{c}_\ell + \alpha$.

$$
\begin{aligned}
\mathbf{y}_\ell &= \text{LayerNorm}\left(\mathbf{x}_\ell + \text{Att}(\mathbf{x}_\ell)\right) \\
\mathbf{x}_{\ell+1} &= \text{LayerNorm}\left(\mathbf{y}_\ell + \text{MLP}(\mathbf{y}_\ell)\right) + \mathbf{c}_\ell \cdot \alpha
\end{aligned}
\tag{1}
$$

$\mathbf{x}_\ell$ is the hidden dimension activation vector at the $\ell$'th layer, $\mathbf{y}_\ell$ the activations after the scaled dot-product attention-mechanism and $\mathbf{x}_{\ell+1}$ the activations after the MLP transformation.

Recent work has illustrated the efficacy of analyzing and manipulating the residual stream by e.g. using sparse autoencoders to extract features from residual stream activations (Huben et al., 2023). This work suggests that the residual stream is a better level of analysis than single neurons (or cohorts of neurons) due partly to the issue of polysemanticity. Targeting the residual stream has also been proposed as a solution to *cross-layer superposition*: the notion that features are smeared across many hidden layers of a deep neural network (Bricken et al., 2023; Templeton et al., 2024). We extract the representation after each layer at the final token of a task example. From these extracted activations we derive layer-specific control vectors. Applying the control vector is the simple addition of adding the $\mathbf{c}_\ell \cdot \alpha$ term to the standard transformer operation as illustrated in Equation 1.

## 2.1 CONTROL VECTORS

A control vector can be trained in multiple ways based on the extracted model activations. The simplest way is by creating a *reading vector*, which is merely the average over the extracted activations. Given a set of prompts $P$, a model with $L$ layers, and $H_\ell(P_i)$ as the hidden state of the residual stream for the $\ell$'th layer and the $i$'th prompt, the reading vector is described as follows:

$$
\mathbf{c}_\ell = \frac{1}{|P|} \sum_{i=1}^{|P|} H_\ell(P_i)
\tag{2}
$$

Related work indicates that for extracting a desired behavior from an LLM it is best to use contrastive pairs. The use of contrastive pairs entails looking at the representations for *positive* and *negative* prompts. For the case of say, emotional valence, it could entail examples of 'happy' prompts and examples of 'sad' prompts. For the case of reasoning we look at examples of successful reasoning and unsuccessful reasoning – see subsection 3.1. This ensures that the control vectors are based on a difference in representations (Zou et al., 2023). The control vector can thus be scaled to induce the desired behavior or its opposite. Given a set of contrastive pairs of positive and negative prompts $P^{\pm} = (P^+, P^-)$, we define the control vector as:

$$
\mathbf{c}_\ell = \frac{1}{|P^{\pm}|} \sum_{i=1}^{|P^{\pm}|} \left( H_\ell\left(P_i^+\right) - H_\ell\left(P_i^-\right) \right)
\tag{3}
$$

Another method for training control vectors involves applying Principal Component Analysis (PCA) to the contrastive activations (Zou et al., 2023). PCA finds directions in the data that account for the most variance, with the first component indicating the *direction* in the data explaining maximum variance. The intuition is that the difference between these contrasting representations will contribute significantly to the variance in the activations, and the first principal component should thus approximate the direction in the activation space that most effectively distinguishes between the desired outcome and its contrast. This direction can then be used as the control vector. Similar to reading vectors using contrastive pairs the PCA control vector is defined in the following manner.

$$
\mathbf{c}_\ell = \text{PCA}\left(\left\{ H_\ell\left(P_i^+\right) - H_\ell\left(P_i^-\right) \quad \forall\, i \in \left[1, \ldots, |P^{\pm}|\right] \right\}\right)_{(1)}
\tag{4}
$$

A key consideration is how much one wishes to modulate representations (controlled by $\alpha$), as too strong modulation could result in non-sensical outputs. When creating control vectors of the *reading*

type, an $\alpha = 1$ is equivalent to adding a "full" activation vector to the signal as $||\mathbf{c}_\ell|| \simeq ||H_\ell||$ which follows from Equation 2. However, when applying PCA we get $||\mathbf{c}_\ell|| = 1$. To account for the difference we implement an up-scaling of PCA-based control vectors which is dependent on the actual norm of the extracted residual stream activations (see subsection A.1).

The derivation of control vectors is thus a relatively simple process that does not induce a large computational overhead. We simply need to extract activations while performing inference on a given task to control and improve the outputs of LLMs without any further training.

## 3   EVALUATION & ANALYSIS

### 3.1   EXPERIMENT

For IOI we create $2,000$ examples for each condition of the following form:

> **Mary**$_{[A]}$ and **John**$_{[B]}$ went to the *store*. **John**$_{[B]}$ gave the *groceries* to **Mary**$_{[A]}$.

This task is inductive as multiple answers could be correct, we are however interested in controlling the model to generate a desired name and pose this as an inductive inference problem. We study four different conditions: ABBA (A), BABA (B), ABBA-Long (AL) and BABA-Long (BL).

bAbI comprises various reasoning tasks, one related to deductive reasoning of which there are $2,000$ examples. The questions have the following format[2]:

> **Passage:** Mice are afraid of wolves. Gertrude is a mouse. Cats are afraid of sheep.
> **Question:** What is Gertrude afraid of? **Answer:** Wolf

GSM8K consists of high quality grade school math problems on which relatively capable LLMs still struggle. The questions are of the following format:

> **Question**: "Natalia sold clips to 48 of her friends in April, and then she sold half as many clips in May. How many clips did Natalia sell altogether in April and May?" **Answer**: "Natalia sold $(48/2 = 48/2 = 24)$ 24 clips in May. Natalia sold $(48 + 24 = 48 + 24 = 72)$ 72 clips altogether in April and May. #### 72"

We provide a fully detailed example of a question from the GSM8K dataset in subsection A.3. For GSM8K we use a total sample of 400 prompts to derive the control vectors. For each dataset we create train and test splits with stratified labels. We then derive the control vector based on model representations when it generates outputs on examples from the train split and test model performance with a control vector applied on the test set.

In order to produce the contrastive pairs we need positive and negative prompt examples. For the positive examples, an obvious method might be to select all cases where the model successfully solves a task, making sure that the class labels are balanced so as to ensure that the model is not biased towards a specific answer. A prompt for unsuccessful reasoning is less clear however. We propose multiple schemes to elicit representations typical of *poor reasoning*. 1) A naive approach entails asking a model to produce an incorrect answer. This was found during testing to be a poor scheme since producing the wrong answer when prompted to might be considered *good* reasoning. 2) Taking examples where the model answers a question incorrectly, this approach does however have pitfalls. In some cases an answer is not actually wrong, but simply not the correct token we were envisioning. As an example in the IOI task "***Mary***$_{[A]}$ *and* ***John***$_{[B]}$ *went to the store.* ***John***$_{[B]}$ *gave the groceries to* ***Carl***$_{[C]}$." Carl is not necessarily wrong, it is just not the token we expected. 3) The third scheme consists of simply creating negative prompts that are 75 random character strings sampled from A to z. While this scheme does not introduce a natural contrast it provides "a point of reference" in terms of model representations and empirically works better than merely using *reading vectors*. We test scheme 2) and 3) in the following experiments.

We carry out multiple experiments applying the intervention to the residual stream of various LLMs at the middle layer at varying scales of $\alpha$ and assess how this influences performance on the three

---

[2]Questions are generally more complicated than this example. See subsection A.2 for more details.

datasets. When deriving control vectors we get a control vector for each layer, previous work however indicates that only applying the vectors to the middle layer is enough to induce strong changes to model outputs (Templeton et al., 2024). We generally assess the impact of the intervention at $\alpha \in [-1, 1]$ at increments of $0.1$, but look at a range of $[-3, 3]$ for Mistral-7B-Instruct.

## 3.2 EVALUATION METRICS

We benchmark performance by investigating the output logits of a constrained set of potential responses, where the potential responses are the collection of all answers in the dataset, similarly to what is done in the *lm-eval* framework (Sutawika et al., 2024). We do not use an exact-match metric as it is particularly error-prone in 0-shot prompt scenarios for smaller LLMs, where the desired structure of the output is unknown to the model. This results in verbose and prosaic language generated by the model to potentially be labeled as inaccurate despite providing correct answers. This issue is less pronounced in few-shot prompting.

We thus use few-shot examples of the task when extracting the activations, in part to alleviate the above-stated issues, but more importantly because LLMs have been shown to be in-context learners (Brown et al., 2020). We furthermore cannot expect to be able to improve a model on a task it cannot adequately solve when the method of improving a model is derived directly from it's own hidden states. For the GSM8K task the model generates a full response with a reasoning trace and we use a decoding scheme to extract the final answer.

Beyond benchmarking the model outputs, we aim to understand the distribution changes as control vectors are applied. We analyze this by examining summary statistics of the resulting distributions and comparing them to the original model. We specifically focus on Kullback-Leibler (KL) divergence between the logit distribution of a model with and without the intervention applied at various $\alpha$ levels, providing a metric for distribution change magnitude:

$$D_{\mathrm{KL}}(P \| P_\alpha) = \sum_{x \in \mathcal{X}} P(x) \log \left( \frac{P(x)}{P_\alpha(x)} \right) \tag{5}$$

where $P$ is the original logit distribution and $P_\alpha$ is the modified logit distribution. We additionally look at model entropy as a function of $\alpha$, revealing whether the model becomes more or less confident with increasing $\alpha$:

$$H(X) = -\sum_{x \in \mathcal{X}} P_\alpha(x) \log P_\alpha(x) \tag{6}$$

Finally, we assess the average probability for correct versus incorrect answers as a function of $\alpha$. This is done to assess how the intervention concretely influences the token probability distribution.

$$\bar{P}_\alpha^{\mathrm{correct}} = \frac{1}{N} \sum_{i=1}^{N} P_\alpha(\hat{y}_i | x_i) \tag{7}$$

$$\bar{P}_\alpha^{\mathrm{incorrect}} = 1 - \bar{P}_\alpha^{\mathrm{correct}} \tag{8}$$

where $N$ is the number of samples, $\hat{y}$ is the predicted answer to the input $x_i$, and where $\hat{y}$ is constrained to the list of potential answers.

These metrics yield insights into how the control vector intervention affects model states. If a certain level of control creates a disproportionate spike in the KL divergence it would suggest a certain threshold in terms of bringing the model out of its usual state. We expect a linear increase in KL divergence as $\alpha$ increases. Entropy indicates the *uncertainty* contained in a probability distribution; we expect that if the control vector improves task accuracy the entropy should drop as the probability mass concentrates on the correct answer token. Finally, assessing the probability mass concentrated at the correct token tells us concretely whether we are generally (across many samples) improving the ability to correctly solve the task.

If we observe decreasing entropy and increased probability on correct tokens as accuracy increases this suggests that the control vector intervention is working as intended. However, if the metrics do not align and we observe for instance decreased entropy but no increase in accuracy it might indicate that we are affecting the model in unintended ways.

We do **not** train any models, but extract control vectors based on hidden states of the models. We evaluate model performance with the intervention on the test set and derive the control vectors from activations based on the training set. We do this to avoid data contamination although we cannot be certain that the training data of the models we use have not been contaminated indirectly.

## 3.3 MODELS

We work with models from the Pythia suite developed by EleutherAI (Biderman et al., 2023). We specifically look at two models, namely Pythia-1.4B and Pythia-2.8B. We analyze these models based on initial benchmarking studies and because this range allows us to analyze the efficacy of the representation engineering approach across different scales. We additionally test the approach on Mistral-7B-Instruct to scale the approach to a larger more capable model and to assess how the control vectors affect an instruction-fine-tuned model.

## 3.4 RESULTS

In this section we report results for *Pythia-1.4B*, *Pythia-2.8B* and *Mistral-7B-Instruct* with the PCA derived control vectors applied. We generally see improved performance across models and tasks, although there are differences in the optimal value of $\alpha$ as well as whether to scale the control vector negatively or positively.

### PYTHIA

We train control vectors on the signal extracted from the **A** condition and apply them across experimental conditions. We observe slight generalization across task variations for both Pythia-1.4B and Pythia-2.8B as seen in 2a and 9a in subsection A.4 for Pythia-1.4B. Pythia-2.8B improves accuracy slightly when the control vector is applied across conditions, with some variation in $\alpha$, and KL divergence increases quadratically as $\alpha$ is increased in both directions, see 2b. The intervention is mostly effective for B and BL suggesting that the control vectors are encoding position-sensitive information of the indirect-object to be generated. These findings are corroborated by the entropy analysis which indicates a slight decrease as accuracy increases, however, we observe that as entropy decreases KL divergence increases. We furthermore see that the probability mass on average accumulates slightly around the correct answer with an increase in $\alpha$ in 2d although the average change in probability on the correct token is minute. We report only on the IOI task for Pythia models as they were not capable of adequately solving neither the bAbI or GSM8K tasks.

### MISTRAL

We see similar results for Mistral as for Pythia-2.8B on bAbI, see Figure 3. We improve the logit-based accuracy and KL divergence increases quadratically. We do, however, observe a slight upward trend in entropy against our expectations of how the intervention should affect the final logit distribution when it successfully improves a model's accuracy. We assess performance on the bAbI dataset as IOI is too simple relative to the performance of the model.

We additionally find that the control vector intervention improves model performance on the GSM8K task; the approach successfully improves the model's ability to solve the task with a negative $\alpha$ as indicated by Figure 4. While model representations are not very robust to the intervention (based on the jagged trend line), we do find that the intervention works. The KL divergence and entropy measures look quite different for GSM8K, although there does seem to be a positive correlation between the metrics.

Finally we assess how control vectors derived from the bAbI task influences performance on GSM8K evaluation and vice-versa. In Figure 5 we report the results of of this experiment. We find an almost identical increase in accuracy on the GSM8K test set when a bAbI control vector is applied, and see similar trends in the KL divergence and entropy plots. We also find that a GSM8K control vector improves performance on the bAbI task, see subsection A.5. This suggests that the representation from which we derive the control vector captures aspects of the information-processing a model performs when solving a reasoning benchmark task.

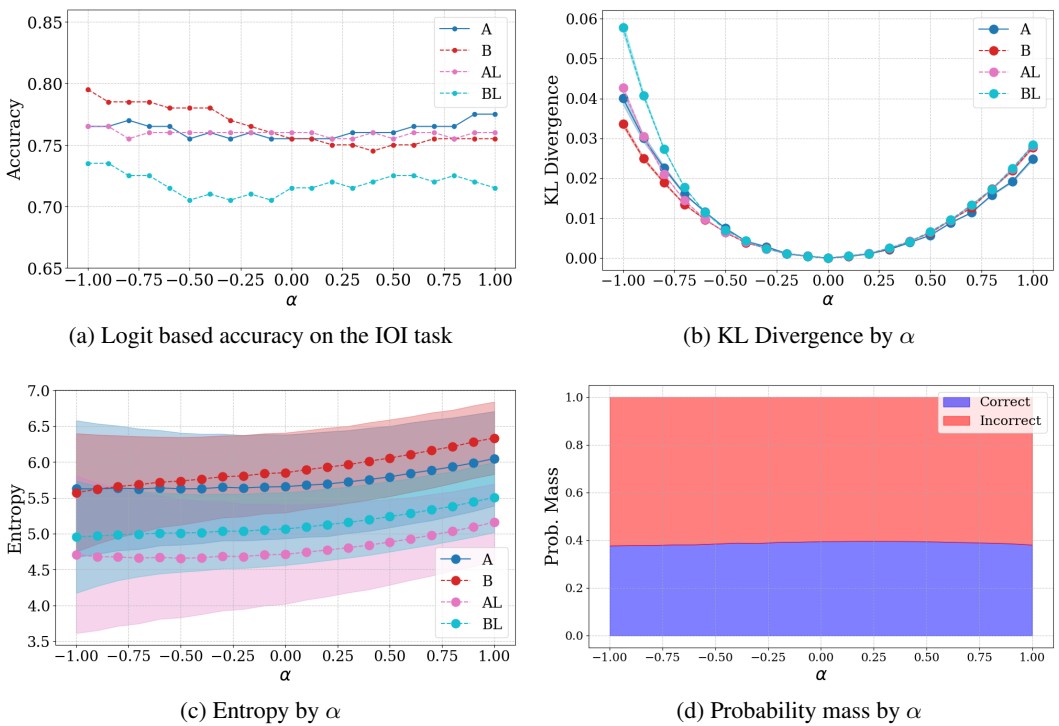

(a) Logit based accuracy on the IOI task

(b) KL Divergence by $\alpha$

(c) Entropy by $\alpha$

(d) Probability mass by $\alpha$

Figure 2: Pythia-2.8B: results from PCA derived control vectors on the IOI task. We observe slight improvement across most conditions and more stable model representations than for the smaller model.

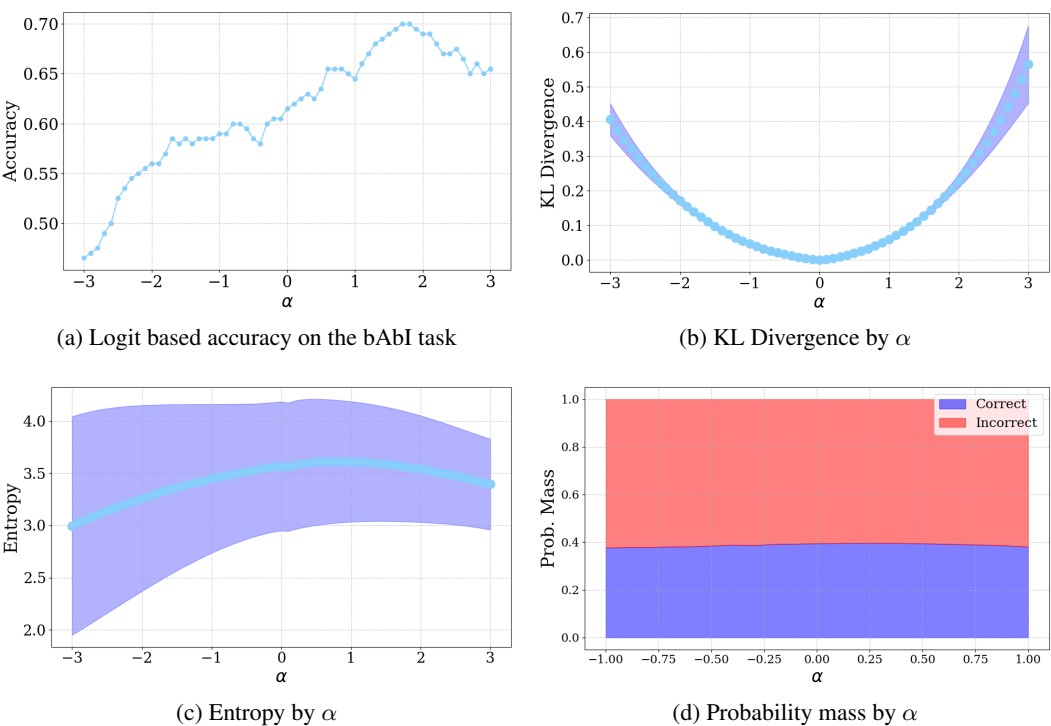

(a) Logit based accuracy on the bAbI task

(b) KL Divergence by $\alpha$

(c) Entropy by $\alpha$

(d) Probability mass by $\alpha$

Figure 3: Mistral-7B-Instruct: results from contrast-based control vectors on the deductive bAbI dataset.

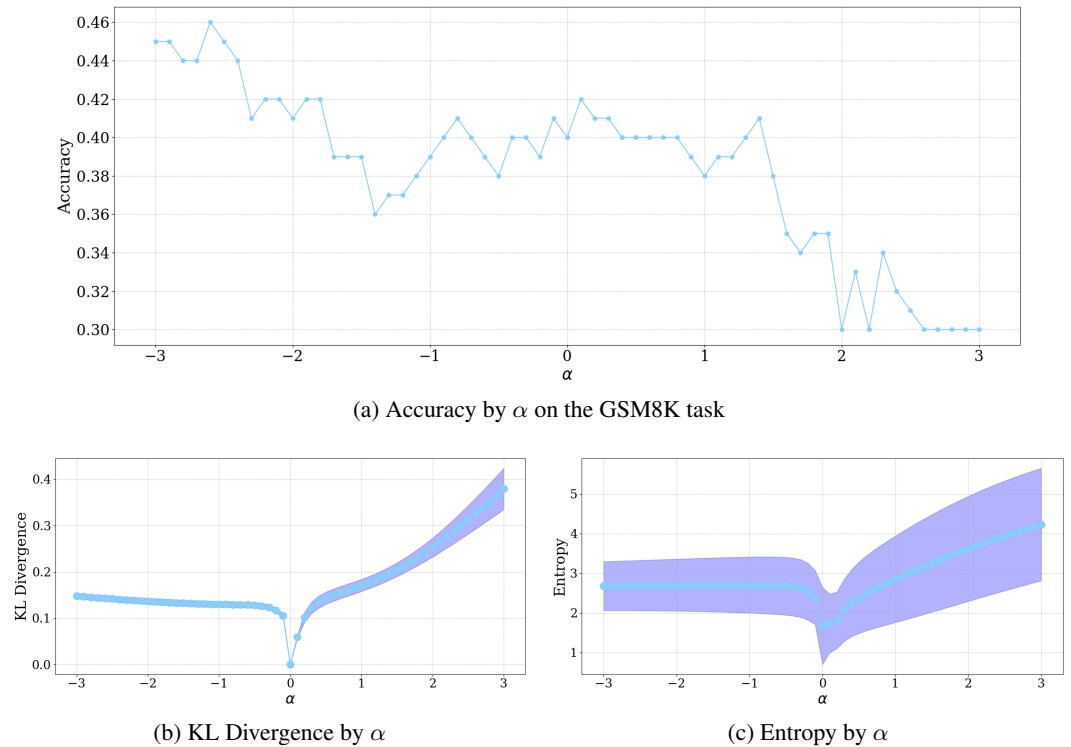

Figure 4: Mistral-7B-Instruct: we report the changes in model performance on the GSM8K task when we apply control vectors derived from the same task.

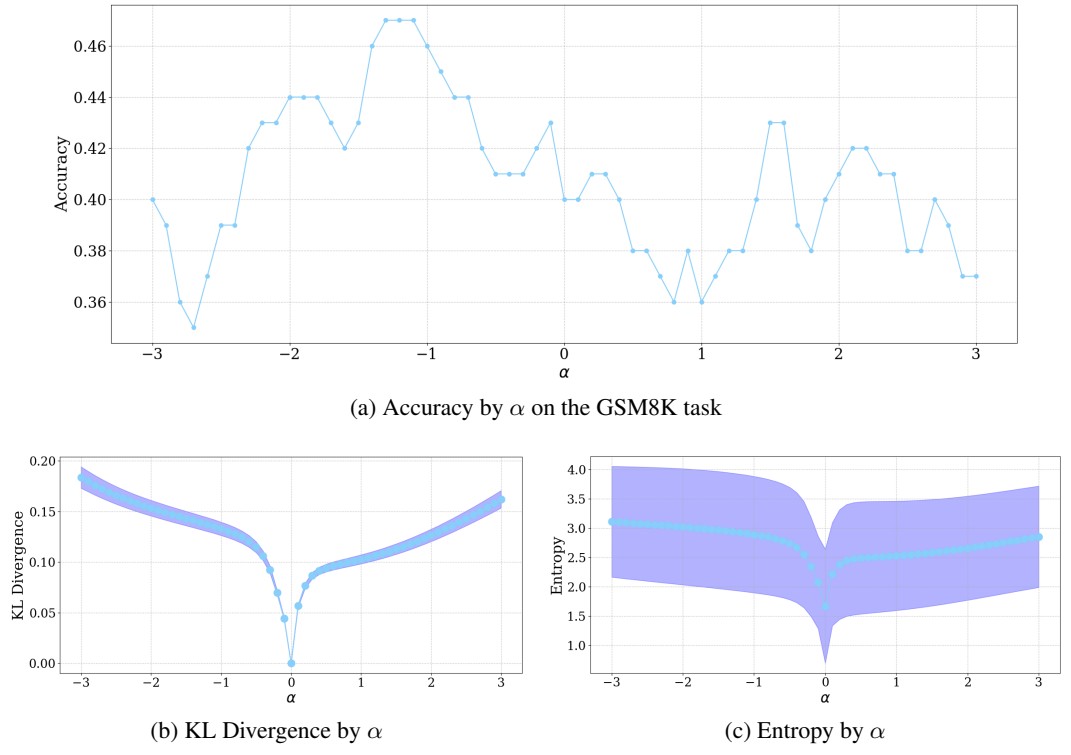

Figure 5: Mistral-7B-Instruct: we report changes in model performance on the GSM8K task when we apply control vectors derived from the bAbI task at varying degrees of $\alpha$. Control vectors derived from bAbI can improve the model's accuracy on the GSM8K task.

**Question**

A robe takes 2 bolts of blue fiber and half that much white fiber. How many bolts in total does it take?

| 0 | -2.6 |
|---|---|
| (...) 1. We know that the robe takes 2 bolts of blue fiber. 2. We also know that it takes half that much white fiber. 3. So, to find the total number of bolts required, we can add the blue and white fiber together: Total bolts = Blue fiber + White fiber. Total bolts = 2 + (1/2) Total bolts = 2.5 Therefore, the robe takes a total of **2.5 bolts**. | (...) 1. We know that the robe takes 2 bolts of blue fiber. 2. We also know that it takes half that much white fiber, which is 1 bolt of white fiber. 3. To find the total number of bolts required, we simply add the amount of blue and white fiber: Total bolts = Blue fiber + White fiber. Total bolts = 2 + 1. Total bolts = 3. Therefore, it takes **3 bolts** of fiber in total to make the robe. |

**Question**

A candle melts by 2 centimeters every hour that it burns. How many centimeters shorter will a candle be after burning from 1:00 PM to 5:00 PM?

| 0 | -2.6 |
|---|---|
| (...) to calculate the total amount of time the candle burns and then multiply that by the rate at which it melts.\n\n1. The candle burns for 5 hours (from 1:00 PM to 5:00 PM).\n2. During this time, it melts 2 centimeters per hour.\n3. So, the total amount of melting is 5 hours x 2 centimeters/hour = 10 centimeters. \n\nTherefore, a candle will be **10 centimeters** shorter after burning from 1:00 PM to 5:00 PM. | (...) to calculate the total number of hours the candle burns and then multiply that by the rate at which it melts.\n\n1. The candle burns from 1:00 PM to 5:00 PM, which is a total of 4 hours.\n2. The candle melts by 2 centimeters every hour, so over 4 hours, it will melt by 2 x 4 = 8 centimeters.\n\nTherefore, the candle will be **8 centimeters** shorter after burning from 1:00 PM to 5:00 PM. |

Figure 6: Full responses created by Mistral-7B-Instruct at $\alpha = 0$ and $\alpha = -2.6$. The model seemingly changes its reasoning trace to reach the correct answer.

We additionally analyze how model responses change when the intervention is applied with the optimal $\alpha$ for a specific model and task. We report two examples in Figure 6 that showcase how Mistral-7B-Instruct generates a different reasoning trace that results in the correct answer being generated as opposed to when no control is applied (see also Figure 8 in subsection A.3 for an incorrect example). More generally, we observe that differently sized models are affected differently by different levels of $\alpha$. Whether this is because of larger models being more robust to changes in their representations or merely the fact that only modifying the middle layer representations is a smaller part of a larger model is an open question.

In summary we find that control vectors can successfully be used to modify the representation of LLMs to improve performance on the specified tasks for all models, although to varying degrees. The finding that the intervention works for a task as complicated as GSM8K is especially affirming, and even more so that it works across tasks. The qualitative examples showcase that applying the interventions retains a model's ability to generate coherent text and that it is not merely inducing the generation of a specific subset of tokens; an important result for the intervention to be applicable outside of research, which is largely focused on improving performance on benchmark datasets.

## 4 DISCUSSION

In this paper, we study the ability of LLMs to solve reasoning tasks by modeling 'reasoning' as a direction in the residual stream. Fundamentally, training control vectors relies on the representations learned by an LLM and the ability to modify these in a way that preserves a model's ability to generate semantically meaningful text. Similar approaches have been applied to modify the "sentiment" of model outputs while maintaining semantic coherency. This research has especially focused on inducing feelings of positivity and negativity effectively creating control vectors equivalent to prepending a prompt with "Pretend you are feeling [X]" (Liu et al., 2023; Hendel et al., 2023). We have similarly shown that we are capable of modulating the representational space of LLMs to boost performance on a simple task while maintaining the model's fundamental ability to generate coherent text. This finding hints that the "ability" of LLMs to perform well on reasoning tasks is encoded similarly to other model states, such as generating semantically positive or negative outputs. Given prior literature, our ability to modify the outputs of an LLM via the residual stream is in and of itself not a surprising finding; the main contribution of the paper is that we can improve on reasoning tasks of varying complexity by nudging a model towards a specific state.

### 4.1 IMPLICATIONS

While there are many questions to be answered regarding reasoning in LLMs and many open questions wrt. to our findings and experiments, they suggest that reasoning performance can be modulated similarly to how we can modulate the emotional valence of generated text or a model's ability to play chess. The developments over the past couple of years have been impressive, and they have

affected the beliefs within the field regarding the limitations of machine learning models trained solely on text. Whether current model architectures are at all capable of anything like the System 2 thinking done by humans seems unlikely, although work is being done to make LLMs reason "properly".[3] Whether such approaches amounts to reasoning in the traditional sense is still up for debate, especially as *reasoning* is a hard term to ground and define.

## 4.2 REASONING

While we acknowledge the deep philosophical questions surrounding the nature of reasoning, our work takes a deliberately narrow empirical approach. We investigate whether specific manipulations of model representations can improve performance on tasks that require reasoning-like behaviors, without making strong claims about the nature of reasoning itself. We do not attempt to define reasoning. Instead, we are interested in using tasks that are recognized and categorized as being related to reasoning, be it inductive, deductive, mathematical, or in theory anything else. These tasks differ in their prompt lengths and in how surface-level the correct solutions are (i.e, whether the solutions requires reasoning through intermediate steps or calculations). In the IOI task, the answer is present in the text, but needs to be inductively recalled, in the bAbI tasks the answers are also in the prompt, but requires computing through one or more intermediate steps to achieve the answer. Finally, in GSM8K, the answers are not present in the text anymore, and they also require parsing the results through multiple intermediate calculations. Through this exploration, we aim to identify actionable strategies for enhancing model performance across a spectrum of reasoning tasks of scaling complexity, irrespective of how reasoning is formally defined.

## 4.3 LIMITATIONS

While this paper showcases how representation engineering can successfully be used to induce better reasoning performance on an inductive, deductive, and a mathematical task, we see certain limitations. We assess a limited scope of models, models that are smaller than production-level state-of-the-art models. Working with smaller models has however provided the opportunity to assess the "emergence" of capabilities and symmetries as models are scaled. Working with models that were developed specifically for interpretability research furthermore opens the door for future research into the internal model dynamics related to LLM reasoning. And while the biggest model comprises a mere 7 billion parameters, Mistral-7B-Instruct is relatively capable.

The IOI reasoning task is furthermore a relatively simple task, which has both advantages and disadvantages. The results indicate that a task such as IOI may even be *too* simple to elicit a general representation. The task is mostly interesting to our research when models cannot correctly answer every question, a hard balance to strike. Hence the switch to the bAbI and GSM8K datasets, and the Mistral-7B-Instruct model for additional experimentation. However, a simple task also limits the scope of potential errors, which results in a cleaner picture of model behavior.

We furthermore suggest first steps towards creating contrastive pairs for reasoning to be used in representation engineering research, we however call for future work to focus on understanding how to optimally derive control vectors from model representations.

## 4.4 CONCLUSION

We successfully used the technique of representation engineering by first training control vectors on a training set of reasoning related tasks and secondly intervening on the middle layer of the residual stream during inference. This allowed us to improve the performance of Pythia and Mistral models on a simple inductive task (IOI), the deductive bAbI task, and finally the GSM8K dataset. We illustrate how control vectors, trained using contrastive prompts, can be used to control and direct the output of LLMs, as well as how these modifications affect the internal representations of these models. Future research directions entail applying the intervention to much larger models, to more general tasks, to test different contrastive schemes as well as exploring ways of embedding this methodology into the models. In conclusion, our experiments indicate that reasoning can in part be understood to be encoded in the residual stream, similarly to how emotional valence or the *Golden Gate* feature is encoded in the models developed by Anthropic (Templeton et al., 2024).

---

[3]see e.g. the most recent models from OpenAI that focus primarily on reasoning (OpenAI, 2024).

## STATEMENT OF REPRODUCIBILITY

We are committed to aiding reproducibility of our work and encourage further research in this direction. The source code for our implementation, developed as an internal framework, will be published alongside this paper to facilitate this process. Our framework is built as a wrapper around PyTorch, enabling easy extraction of hidden dimension representations and application of control vectors. The models described in section 3.3 were loaded using the HuggingFace API and details on model version are described there. The training process for control vectors is detailed in section 2.1. The calculation of experimental results, including all metrics used, is thoroughly described in section 3.2. Information about datasets used, including any preprocessing steps, can be found in section 3.1 and A.2.

Our code is publicly available through Github, along with documentation to guide users through the setup and execution of experiments.

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

# A APPENDIX

## A.1 CONTROL VECTOR NORMS

As can be gathered from equation 2 any control vector will have a norm that is equivalent to the average of the hidden dimension activations at a given layer of the model. For the contrastive pairs (equation 3) it will be equivalent to the norm one gets when subtracting the "opposite" state from the "desired" state. This is however not the case for the control vector based on a PCA, because a standard PCA implementation scales the output vector to have an L2 norm of 1. We cannot know the exact size of $||H_\ell||$ as this is dependent on the learned parameters $\gamma$ and $\beta$ of the LayerNormfunction. Assuming $\gamma = 1$ and $\beta = 0$, which are the usual initial values (Ba et al., 2016), a rough estimate of the norm of the hidden dimension, d, is $||H_\ell|| = \sqrt{d}$, the value however varies a lot dependent on the model and between layers. We therefore scale the PCA control vectors by $||H_\ell||$ of the training data to allow for a more direct comparison between the control vector methods. As indicated by figure 7 we get identical norms for the two methods, while the actual control vectors are different.

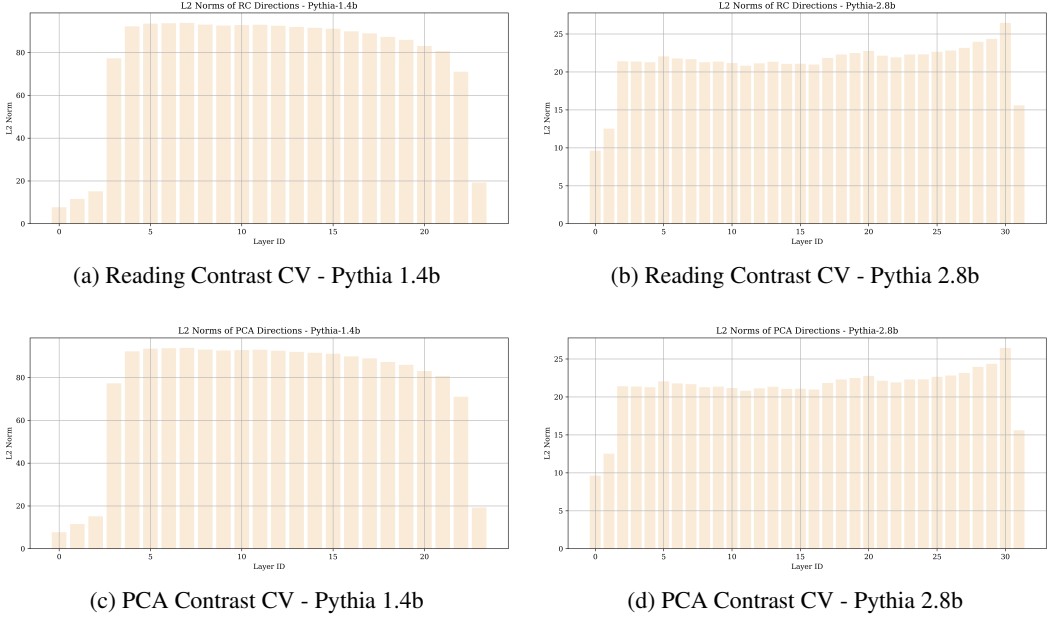

(a) Reading Contrast CV - Pythia 1.4b      (b) Reading Contrast CV - Pythia 2.8b

(c) PCA Contrast CV - Pythia 1.4b      (d) PCA Contrast CV - Pythia 2.8b

Figure 7: Control Vector Norms

## A.2 DATA

### INDIRECT-OBJECT-IDENTIFICATION (IOI)

The IOI task examples were generated using the following schema:

> [A] and [B] went to the [location]. [B] gave the [object] to [A].

In the BABA (**B**) condition the schema was:

> [B] and [A] went to the [location]. [B] gave the [object] to [A].

We provide an example here, also shown in 3.1.

> **Mary**[A] and **John**[B] went to the *store*. **John**[B] gave the *groceries* to **Mary**[A].

Examples are split into $(X, y)$ pairs of the following form and fed to the model with X as the input and y as the correct generation:

("*Mary and John went to the store. John gave the groceries to*", " **Mary**")

When extracting model representations (X, y) pairs are combined into a single string containing the input prompt and the correct answer. When evaluating the performance of the derived control vector the models only receives X, and we evaluate whether the token with the highest logit of the potential answers is equivalent to the correct label, y. We used an 80/20 train-test split to train and evaluate the performance on control vectors. We do not perform any additional pre-processing.

### BABI DATASET

We used the bAbI dataset, focusing on task 15, which involves simple deductive reasoning. Each example in the dataset consists of: 1) A passage containing factual statements, 2) a question based on the passage and 3) the correct answer. We provide an example here.

> **Passage:** Mice are afraid of wolves. Gertrude is a mouse. Cats are afraid of sheep. Winona is a mouse. Sheep are afraid of wolves. Wolves are afraid of cats. Emily is a mouse. Jessica is a wolf.
> **Question:** What is Gertrude afraid of?
> **Answer:** Wolf

Examples are split into $(X, y)$ pairs of the following form and fed to the model with X as the input and y as the correct generation:

> ("*Passage:\nCats are afraid of wolves.\nGertrude is a mouse.\nJessica is a cat.\nQuestion: What is Jessica afraid of?\n*", " **wolf**")

When extracting model representations (X, y) pairs are combined into a single string containing the input prompt and the correct answer. When evaluating the performance of the derived control vector the models only receives X, and we evaluate whether the token with the highest logit of the potential answers is equivalent to the correct label, y. We used an 80/20 train-test split to train and evaluate the performance on control vectors.

We do not perform any additional pre-processing and have downloaded the data directly from `https://huggingface.co/datasets/Muennighoff/babi`.

### 1-SHOT PROMPTING

When 1-shot prompting, we simply provide the input prompt with an example of how to solve the task. The model is prompted with an out-of-distribution example in order not to nudge the model towards a specific answer. This takes the following form:

> Example\nPassage: Giraffes are afraid of lions.\nElephants are afraid of snakes.\nJohn is a giraffe.\nQuestion: What is John afraid of?\nAnswer: Lion\n*(Passage and question)*

### INSTRUCTION PROMPTING

Since instruction fine tuned models are able to follow instructions, we found that prompting the model to output a one word answer made sense, and was also found to work better empirically.

> [INST]\n*(potential one-shot example)*\nPlease give a one word answer as in the example above. [/INST]\n*(Passage and question)*

### A.3 GSM8K FLIPPED RESPONSE

The following is a concrete example from the experiment with Mistral-7B-Instruct and the PCA-contrast control vector with $\alpha = -2.6$. This example showcases how the reasoning trace is still interact when the intervention is applied, and that something in the intervention leads the model to accurately end up at the correct answer of 3 as opposed to 2.5. The "reasoning" in the two samples is identical until the point where the model generates the text "which is 1 bolt of white fiber". This additional information seemingly allows the model to generate the correct response when finally answering the question.

Response before intervention:

> **Question**: "A robe takes 2 bolts of blue fiber and half that much white fiber. How many bolts in total does it take?"
>
> **Correct Answer**: 3.0
>
> **Predicted Answer**: 2.5
>
> **Full Response**: " solve this problem, we need to add the amount of blue and white fiber required for the robe. 1. We know that the robe takes 2 bolts of blue fiber. 2. We also know that it takes half that much white fiber. 3. So, to find the total number of bolts required, we can add the blue and white fiber together: Total bolts = Blue fiber + White fiber. Total bolts = 2 + (1/2) Total bolts = 2.5 Therefore, the robe takes a total of 2.5 bolts."

Response after intervention:

> **Question**: "A robe takes 2 bolts of blue fiber and half that much white fiber. How many bolts in total does it take?"
>
> **Correct Answer**: 3.0
>
> **Predicted Answer**: 3.0
>
> **Full Response**: " solve this problem, we need to add the amount of blue and white fiber required for the robe. 1. We know that the robe takes 2 bolts of blue fiber. 2. We also know that it takes half that much white fiber, which is 1 bolt of white fiber. 3. To find the total number of bolts required, we simply add the amount of blue and white fiber: Total bolts = Blue fiber + White fiber. Total bolts = 2 + 1. Total bolts = 3. Therefore, it takes 3 bolts of fiber in total to make the robe."

**Question**

A new program had 60 downloads in the first month. The number of downloads in the second month was three times as many as the downloads in the first month, but then reduced by 30% in the third month. How many downloads did the program have total over the three months?

| 0 |
|---|

The number of downloads in the second month was three times as many as the downloads in the first month. So, we multiply 60 by 3: 60 x 3 = 180 downloads. Step 2: Calculate the number of downloads in the third month. The number of downloads in the third month was reduced by 30% in the third month. So, we multiply 180 by 0.7 (which is the decimal equivalent of 30%): 180 x 0.7 = 126 downloads. Step 3: Calculate the total number of downloads over the three months. To find the total number of downloads, we add the number of downloads in each month: 60 + 180 + 126 = 366 downloads. Therefore, the program had a total of 366 downloads over the three months.

| 3 |
|---|

Let's solve this problem step by step: 1. In the first month, the program had 60 downloads. 2. In the second month, the number of downloads was three times as many as the first month, so there were 3 x 60 = 180 downloads.\n3. In the third month, the number of downloads was reduced by 30%, so there were 180 - (0.3 x 180) = 120 downloads. 4. To find the total number of downloads over the three months, we add the number of downloads in each month: 60 + 180 + 120 = 360 downloads. Therefore, the program had a total of 360 downloads over the three months.

Figure 8: Full responses created by Mistral-7B-Instruct at $\alpha = 0$ and $\alpha = 3$. The model seemingly changes it's reasoning trace to answer incorrectly.

## A.4 ADDITIONAL RESULTS

We train control vectors on the signal extracted from the **A** condition and apply across experimental conditions. We observe slight generalization across tasks for both Pythia-1.4B and Pythia-2.8B as seen in 2a and 9a. Pythia-1.4B improves accuracy when the control vector is applied with positive $\alpha$ across conditions, and KL divergence is relatively stable in the positive direction and very unstable in the negative direction resulting in much higher and more varied KL divergence (figure 9b). These findings are corroborated by the entropy analysis which indicates a slight decrease as accuracy increases, but an even stronger drop in entropy as $\alpha \to -1$, suggesting that the intervention is pushing the probability mass towards a specific incorrect token (figure 9c). We furthermore see how the probability mass on average accumulates around the correct answer in figure 9d although the increase with positive $\alpha$ is small.

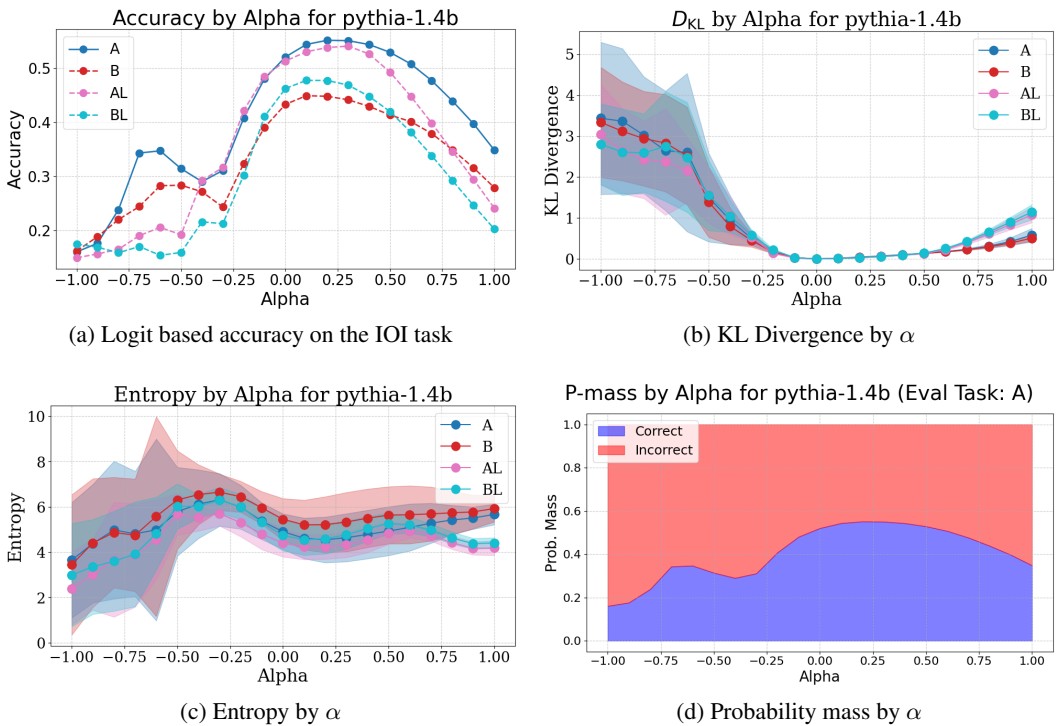

(a) Logit based accuracy on the IOI task

(b) KL Divergence by $\alpha$

(c) Entropy by $\alpha$

(d) Probability mass by $\alpha$

Figure 9: Pythia-1.4B results. Results from PCA derived control vectors. Slight improvements in accuracy across all conditions, and malleable model representations.

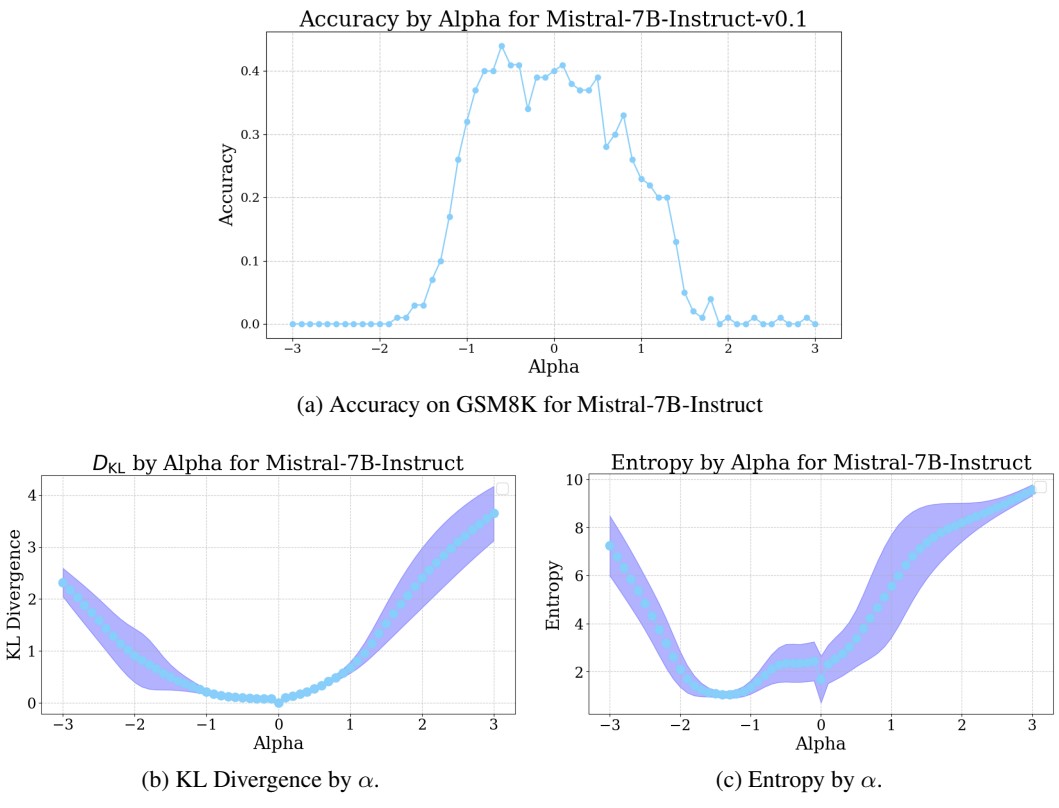

(a) Accuracy on GSM8K for Mistral-7B-Instruct

(b) KL Divergence by $\alpha$.

(c) Entropy by $\alpha$.

Figure 10: Mistral-7B results. Results from PCA-based CVs on **random contrastive pairs**.

## A.5 CONTROL VECTOR CROSS-TASK RESULTS

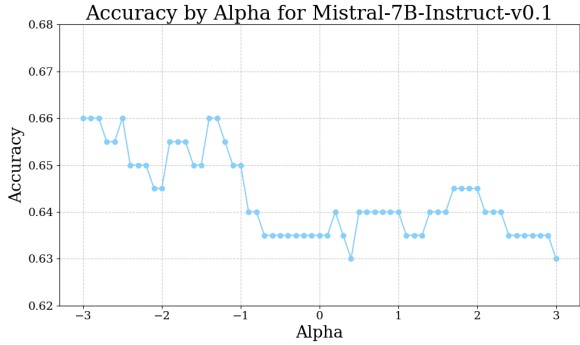

(a) Accuracy on bABI for Mistral-7B-Instruct

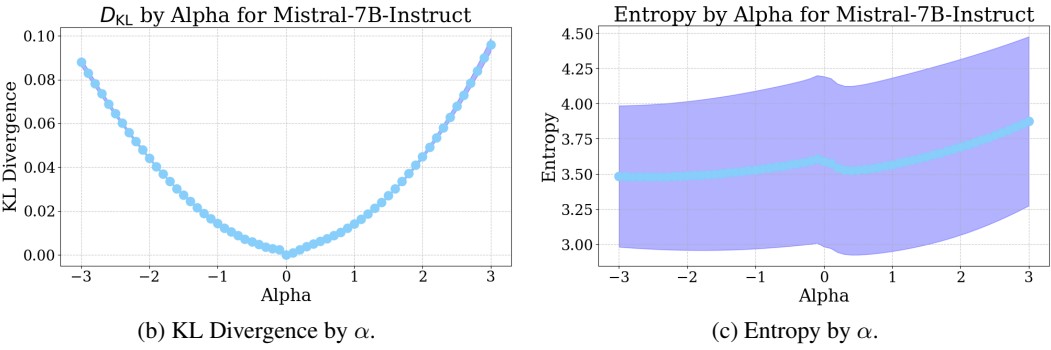

(b) KL Divergence by $\alpha$.          (c) Entropy by $\alpha$.

Figure 11: Mistral-7B results. Results from PCA-based CVs trained on the GSM8K task evaluated on bAbI.

