# OpenReview forum: "Improving Reasoning Performance in Large Language Models via Representation Engineering"
_ICLR.cc/2025/Conference — ICLR 2025 Poster_

### Official Review · Reviewer_TVGr · 2024-10-17

**Soundness:** 2
**Presentation:** 3
**Contribution:** 2
**Rating:** 6
**Confidence:** 4

**Summary:**

This work uses the control vector method to promote task-performant representations in LLMs for IOI and certain bAbI tasks, using a new (but unspecified) software framework, with moderate success. The paper's substantive claims are:

1. These tasks are reflective of LLM reasoning ability
2. The corresponding task-representation vectors in the residual stream are appropriately representative of reasoning in LLMs
3. The observed increase in task-performance after applying the method is evidence for an improvement in reasoning as a broad faculty (as opposed to only the specific task at hand.)
4. There is no intrinsic difference between reasoning and other information-processing tasks as far as an LLM is concerned.

**Strengths:**

It is very clearly presented what the paper wishes to achieve, and the methodology is clear.
The introduction adequately situates this work in the context of prior work in the field.
To the best of my knowledge, the control-vector method has not been tried before on this particular task.
I particularly enjoyed the meticulous introduction that covered the conceptual and mathematical bases for the remainder of the work; it read well as a mini-tutorial on the techniques used here.

**Weaknesses:**

It feels as if this paper wanted to be about a software tool or framework, but "reasoning" was more attractive as a topic.
Of the four substantive claims concerning reasoning, I don't think a single one is substantiated.
These 4 criticisms are the foundations of my current stance on the paper, and I would be happy to raise my score accordingly if these concerns were adequately addressed (by e.g. weakening claims or strengthening their support.)

1. The tasks are reflective of LLM reasoning ability.

That is a stretch, right? IOI is out of the question. bAbI is a sort of edge-case, but consider that Weston et al frame the suite as toy tasks, and that the success of various memorisation architectures on the taskset afterwards have prompted the development of new benchmarks that aren't hackable by exploiting statistical regularities. The claim that this is about "Reasoning" is too heavy for the experimental foundations.

2. The corresponding task-representation vectors in the residual stream are appropriately representative of reasoning in LLMs

This is essentially what the paper needs to establish the existence or nonexistence of empirically; that there exists some broad and task-general faculty in LLMs (that we would call Reasoning), and that this faculty has a manipulable representation in the residual stream. Even if I am willing to take all the literature on residual streams at face value, the onus is still on the authors to establish that the tasks they have chosen are really reflective of Reasoning --- that's claim 1.

3. The observed increase in task-performance after applying the method is evidence for an improvement in reasoning as a broad faculty (as opposed to only the specific task at hand.)

This is crucial. If applying the control vector only improves an LLMs ability to do bAbI tasks, then we're not saying anything about a capacity to Reason: we would instead call it "a capacity to do bAbI", similarly to a "capacity to play chess". If the authors insist on framing the contribution around Reasoning, then at least there ought to be some evidence of generality: if it turns out that boosting performance for bAbI also increases performance on the GSM suite or ARC or anything else, then that would be very interesting to see! Due to the "context-freeness" of Reasoning, there is no way around this particular hurdle apart from empirical demonstration of the efficacy of the technique in general performance boosts across substantially different domains. I can't see any way to get around this purely rhetorically: whatever argument X the authors would like to make that their current experimental basis is sufficient evidence for improvement at a broad capacity for Reasoning, I can take that argument X and apply it for the cited work (line 59) that residual stream modification improves chess and othello performance, undermining the authors' claims to conceptual novelty.

4. There is no intrinsic difference between reasoning and other information-processing tasks as far as an LLM is concerned.

It's unclear that the empirical observations obtained in this work are sufficient to substantiate this claim. It does not help the matter that the authors deliberately avoid taking any stance on characterising what Reasoning might be, or what properties it might or might not have (lines 38, 438), which makes it difficult to evaluate what the claim is even supposed to mean. I would like to view this refusal to take a stance on what-Reasoning-is charitably as scholarly circumspection, but when juxtaposed with unsubstantiated claims about Reasoning ability in LLMs, I'm afraid that citations like Johnson-Laird read more like a checklisting exercise than material engagement with the literature.

**Questions:**

- If you are framing this around "Reasoning", what kinds of philosophical commitments are you making? For example, it seems you implicitly want to frame reasoning as something that cannot be characterised purely behaviouristically, which would necessitate the kind of "going into internal representational structure" that you seem to want to frame the control-vector technique as; so maybe it's worth doubling down on Reasoning as something involving mental states and representations with causal force. However, since you also want to conclude that reasoning is not an innately distinguished ability in LLMs, are you implicitly discounting prompt-level interventions such as CoT and their efficacy at improving task performance? What conception of Reasoning makes it amenable to be improved by residual stream modification but apparently not by structured CoT?

- Why is there nothing about the software? Evidently it was a useful suite of tools to have gotten you through all of the experiments, why say nothing about it?

---

> ### Author Response · Authors · 2024-11-21
>
> We want to thank the reviewer for taking the time to give in-depth feedback highlighting what they see as core issues of the paper. We agree with some of the problems raised, but will also argue that the review overstates some of the key claims of the paper. We have taken measures to rectify some of the critiques in the hope that the reviewer will reconsider their rating. The major change we’ve made to the paper are the following additions:
>
> - We run control vector experiments on GSM8K with Mistral-7B-Instruct.
>
> - We evaluate cross-task efficacy of control vectors for Mistral-7B-Instruct. This means that we apply control vectors trained on the bAbI task when evaluating on GSM8K and vice-versa.
>
> We have also added an additional section in the discussion (4.2) discussing how the specific tasks analyzed relate to the concept of reasoning. Both additional experiments indicate that the control vector intervention works which we believe strengthens our claims and the general results of the paper.
>
> **Key claims**
>
> The review states that we make 4 key claims of which the reviewer believes none are substantiated:
>
> 1. The tasks are reflective of LLM reasoning ability.
>
> 2. The corresponding task-representation vectors in the residual stream are appropriately representative of reasoning in LLMs.
>
> 3. The observed increase in task-performance after applying the method is evidence for an improvement in reasoning as a broad faculty (as opposed to only the specific task at hand.)
>
> 4. There is no intrinsic difference between reasoning and other information-processing tasks as far as an LLM is concerned.
>
> We appreciate the reviewer's careful attention to our claims and would like to clarify our intended scope:
>
> - **Claim 1: tasks are reflective of LLM reasoning ability**
>
> We argue that these represent aspects of LLM reasoning ability, not that they are generally reflective of all LLM reasoning. We agree that the original task choice is quite simple, IOI being simpler than bAbI. We have added a subsection to the discussion (section 4.2) on how these specific tasks relate to reasoning and how GSM8K is perhaps more reflective of reasoning. We also attempt to briefly touch upon the reviewers note that we do not take a stance as to defining reasoning. We take the note that “citations like Johnson-Laird read more like a checklisting exercise” to heart and attempt to rectify this in the added section 4.2, but a topic such as reasoning is incredibly nuanced as the reviewer highlights and we simply do not have the space to completely justify a full discussion of the concept although it is warranted.
>
> The review additionally remarks that "the development of new benchmarks that aren't hackable by exploiting statistical regularities" which seems unsubstantiated. It is quite clear that smaller models can do quite well on very hard benchmark tasks if they are trained on the correct data. See for example the Phi models developed at Microsoft that some argue have essentially been trained on the test-set.
>
> - **Claim 2: task-representation vectors are appropriately representative of reasoning in LLMs**
>
> Although the paper might have made the impression, we do not aim to argue that the control vectors elicited are fundamentally representing a general concept encapsulating all reasoning ability. We state more clearly in the paper revision that they are actually more task-specific than one might have hoped, although our additional experiments actually show that a control vector trained on bAbI also improve performance for the Mistral-7B-Instruct model on GSM8K and vice-versa. While the additional experiments indicate that the control vectors model a more general representation related to reasoning ability (although experiment are limited to bAbI and GSM8K), we are more careful in not overstating the implications of these initial results.
>
> - **Claim 3: improvement in reasoning as a broad faculty**
>
> Thank you for pointing out that our contribution could be interpreted in a way to make this claim. We did not aim to make this claim and therefore do not state that the control vectors induce an improvement in reasoning as a broad faculty, only that we improve performance on the specific reasoning benchmark. The paper does not present the evidence to support the claim stated by the reviewer. While one could argue that the additional results from cross-task application represented in the resubmitted version is partial evidence for this claim, it is by no means conclusive and not something that is stated in the paper. To further avoid this impression, we added section 4.2 to clarify our claims about reasoning.

---

> ### Author Response · Authors · 2024-11-21
>
> - **Claim 4: no intrinsic difference between reasoning and other information-processing**
>
> We can see that there was ambiguity in our argumentation. Our intention is to provide a specific and limited conclusion and have therefore revised our central arguments as follows:
>
> We have revised certain sentences in the paper to remove the ambiguity:
>
> - Lines 28-30: “Our results suggest that there is no intrinsic difference between the process of reasoning and other information-processing tasks performed by LLMs.” —> “Our results suggest that reasoning performance can be modulated in the same manner as other information-processing tasks performed by LLMs …”
>
> - Lines: 482 - 484:  “While there are many questions to be answered regarding reasoning in LLMs and many open questions wrt. to our findings and experiments, it is nonetheless suggestive of the fact that reasoning is not necessarily different from any other form of processing done by LLMs.” —> “While there are many questions to be answered regarding reasoning in LLMs and many open questions wrt. to our findings and experiments, they suggest that reasoning performance can be modulated similarly to how we can modulate the emotional valence of generated text or a model's ability to play chess.”
>
> - Lines 475 - 476: “This finding hints that the ”ability” of LLMs to do reasoning is encoded similarly to other model states, such as generating semantically positive or negative outputs.” —> “This finding hints that the "ability" of LLMs to perform well on reasoning tasks is encoded similarly to other model states, such as generating semantically positive or negative outputs.”
>
> We have thus generally attempted to be more careful in our wording to reduce the ambiguity of the text.
>
> **Questions**
>
> 1. Framing around reasoning
>
> As you point out, we do not properly address the philosophical discussions of reasoning. We focus specifically on performance on tasks that we relate to reasoning. In the added section 4.2 we argue for our approach to reasoning emphasizing an empirical rather than philosophical approach to the subject. The method we present in this paper does not discount the efficacy of CoT prompting. The control vectors are derived from behavioral outputs from the models. The fact that CoT prompting works could actually be considered supportive of the claim that aspects of the ability to solve reasoning tasks is encoded in the residual stream. It would also be interested to see whether chain of thought prompts could be used as a basis for improving our control vectors.
>
> 2. Software
>
> Thank you for pointing out that we can emphasize the software contribution stronger. The tools developed for this work serve primarily to support our investigation of reasoning in LLMs. We plan to release the code to support reproducibility and future research, they are just not the main focus of our contribution.  This project was developed with deep interest in reasoning as a concept, and analyzing reasoning performance empirically is an initial step in this direction. We have rephrased e.g. the abstract to state open-sourcing of code and moved away from discussing it as a framework. We do plan to make it a python package (currently useable with PyTorch), but it is not the main focus of the paper.
>
> We appreciate the reviewer's (TVgr) thorough engagement with our work's theoretical implications. Our additional experiments with GSM8K and cross-task efficacy help clarify the scope of our claims while suggesting interesting directions for future investigation of how reasoning-like behaviors emerge and can be enhanced in language models.  We have attempted to rectify the critiques with additional experiments and to clear up any misconceptions and disagreements with e.g. the additional subsection of the discussion. We hope this improves the reviewers opinion of the paper and invites a reconsideration of the rating.

---

> > ### Comment · Reviewer_TVGr · 2024-11-23
> > **2nd response**
> >
> > In light of the moderations to the claims and the crucial multimodal transference experiment, **I will readjust my overall score to 6, and I intend to advocate on the authors' behalf for acceptance.** However, I reserve the right to reconsider my score again pending discussions with reviewer KzoH, who appears to have concerns about the admissibility of the new empirical evidence that is decisively affecting my judgment.
> >
> > I appreciate your rebuttal. Please allow me to raise some additional concerns/limitations/additional-angles in light of the new evidence. These do not materially affect my current judgement negatively, but perhaps the authors may wish to discuss these points anyway for the benefit of the work.
> >
> > **1. "Superintelligence" hack**
> >
> > It seems unlikely that we can find the reasoning vector, promote its magnitude, and get unlimited returns on reasoning ability; by all of our current understanding, there *must* be limitations to this procedure. They do not have to be explored here, but perhaps understanding the limitations of control vectors as a separate avenue for performance (cf. training, model size, inference time scaling) is future work where others may build on your foundations in this work.
> >
> > **2. "Reasoning" vector or "correctness" vector?**
> >
> > It is of course beyond the scope of reasonable effort in this paper to distinguish the following potentially confounding factor. Let us assume that there is some vector that promotes ability in both bAbi and GSM8k; *how do we know that this vector corresponds to "reasoning" and not, for example, "correctness on publicly available datasets?"* Given the nature and obscure training regimes of LLMs, it's hard to completely discount the alternative possibility. Therefore, though I accept the new experiment as evidence in favour of a general reasoning capability, I believe this evidence must be qualified as partial. Or, more positively phrased, the evidence is suggestive of an initial promising result that will feed into future work.
> >
> > **3. Software angle?**
> >
> > The experimental turnaround is impressive! This is no doubt a combination of hard work and the strength of your software package. Here is a potential angle to showcase the software.
> > - Clearly the strength of the reasoning claim depends on multimodal transference.
> > - Because we cannot exhaust domains, it is the sort of claim where we can only continue to accumulate positive evidence in search of disproof by trying more experiments of the form "boost benchmark X and try to observe improvement in benchmark Y"
> > - Your software is the plug-and-play methodological foundations for this investigative program; independently of whether the *specific* reasoning claims of this paper hold or eventually fall, the questions you raise merit further attention.

---

> > > ### Author Response · Authors · 2024-11-26
> > >
> > > We want to thank reviewer TVGr again for their thorough analysis and additional points. The critiques were very valuable in terms of how the paper is presented and the additional points you mention in your response will be taken into account for future research. Please also note that we have attempted to address the concerns of kzoH in an additional response. Your input, as well as the input from other reviewers, has helped in improving our paper!

---

### Official Review · Reviewer_imCU · 2024-10-26

**Soundness:** 3
**Presentation:** 3
**Contribution:** 3
**Rating:** 8
**Confidence:** 3

**Summary:**

The paper proposes a method to elicit improved reasoning from LLMs at inference time. Similar to inducing writing style or emotional valence, the researchers suggest that reasoning ability is a controllable characteristic which can be suppressed or enhanced.

Using representation engineering, they read residual stream activations from a training set containing positive and negative reasoning trajectories, and these are used to derive a control vector which, colloquially speaking, points in the “better reasoning” direction. In order to improve results, the paper uses contrastive pairs, and further takes only the strongest PCA component of the pairs in order to only select for the “better reasoning dimension”. At inference time, it is applied to the residual stream to improve the model’s reasoning capabilities.

The paper carries out experiments on a bAbI simple deductive reasoning dataset task and the indirect-object-identification (IOI) task. These are applied to two Pythia models and a larger Mistral-7B-Instruct which demonstrate that applying this method yields improved results compared to the baseline (average probability of correct vs. incorrect). It also shows that the model’s entropy and KL divergence are modified due to this intervention as one would expect.

**Strengths:**

Strong points:
The paper provides a clear overview of the motivation and methodology. It is well-structured and self contained.
The paper performs a simple and novel training-free application of an existing method to improve basic capability of LLMs, where it was previously only used for eliciting style modifications. While other papers have shown that, e.g. truthfulness exists as a direction (or control vector) in representation space, it may still be counterintuitive that a capability such as reasoning is also a “direction” in representation space.
The paper uses auxiliary methods (Entropy & KL divergence) to analyze the results of applying this method.
The paper’s most validating result, as mentioned in the results section: an 8% improvement in the bAbi task over baseline.

**Weaknesses:**

Method:
The paper reports a successful application of the method to two datasets. Due to the need for contrastive pairs in order to generate the control vector, it requires positive (good reasoning) and  negative ones (bad reasoning).
The paper discusses 3 schemes for generating the “negative” pairs:
- Asking the model to produce an incorrect answer. This, as the paper notes, is not really a negative example since the model uses correct reasoning when generating an incorrect answer.
- Selecting cases in which model response is not the expected one (which, as the paper notes may still be “not actually wrong” just not the expected answer)
- Selecting a random string of 75 letters as the model’s output reasoning trajectory, which is surely “wrong”. This is pointed out correctly as being ‘unnatural’.

There are a couple of points that are not clarified:
- The paper mentioned testing the latter two schemes, but it is unclear from the results how these are tested. This should be explained in detail (e.g. are they equally mixed as negatives. This requires clarification (or perhaps an ablation study to understand the effects).
  Please provide details on how the different negative example schemes were combined or compared in their experiments.
- Furthermore, without analyzing the errors made by the model, it is not entirely obvious that either of these schemes actually push the model towards correct reasoning. For example, if most errors are “out of distribution” as in the example in the paper: ”Mary[A] and John[B] went to the store. John[B] gave the groceries to Carl[C], merely forcing the model to remain “within” distribution of tokens it’s already seen will already improve its results even if the modified reasoning paths only choose names at random from A and B. Similarly, making sure the model is pushed away from random strings of letters will do the same, and it is hard to say if reasoning has indeed been improved, as in each case (bABi and IOI) there appears to be a natural incorrect token (the “other name”) measuring its probability as an additional measure (as opposed to just the correct token) may help understand these issues better.

Results and analysis:
It is stated that the entropy is expected to decrease as the model veers towards the correct token, concentrating its mass to it. This would mean that it would be negatively correlated with accuracy. However, this does not appear to be the case for Figure 4, the bAbi on Mistral, unless I missed something. In fact the opposite appears to be true. In this plot the entropy also appears to be positively correlated with the probability mass, which requires explanation. If this is indeed the case, please provide a more detailed analysis of the result, and possibly the discuss the correlation between these results.

**Questions:**

The main questions the paper would benefit from answering (following the above discussion) to strengthen the claim:
1. Is the “reasoning” task specific? Does it benefit other tasks?
2. Which types of incorrect reasoning are reduced when using the method?
Either analyzing the errors or showing improved reasoning on a held-out reasoning task (i.e. using an OOD reasoning task) would strengthen the claims appearing in the paper, which can be argued to be at least somewhat related to non-reasoining improvements.

Following the points from above:
- Please explain how the negative sample for the contrastive pairs were constructed or show an ablation study
- Please explain whether the entropy behaves as expected for the Mistral result.

A few more minor points that would improve readability and clarify confusions:
1. Unless I missed something, the paper does not explain how the reading vector (or control vector) is constructed from the latent representation of the examples. Specifically, there are several tokens in each prompt, and it is not clear how these are treated. It is implicitly understood from the notation that this is not per-token, so perhaps the last one is taken. It would be useful if explained explicitly (perhaps even discuss different design choices), possibly by providing psuedocode, formulas or a diagram.
2. The claims are that the results are improved for all tasks but in different directions. This is taken to mean “different sign of alpha” but it is not clear. There appears to be no natural direction, possibly(?) due to the PCA process.
3. The evaluation as scoring from logits makes sense, but it is not explained how these are chosen, and so possibly may be a source of confusion.
3. It is stated that the KL divergence is expected to rise linearly with alpha, and reported that this is indeed the case. Looking at the plots (Figures 3,4) this seems to be quadratic in nature for the larger models, so it rises linearly with the magnitude of alpha, not with alpha.

---

> ### Author Response · Authors · 2024-11-21
>
> We appreciate the reviewer’s (imCU) thorough feedback highlighting both the strengths and limitations of our work. We have made substantial improvements to address these concerns, particularly through additional experiments that strengthen our core claims. The major changes we have made to the paper are the following additions:
>
> - We run control vector experiments on GSM8K with Mistral-7B-Instruct.
>
> - We evaluate cross-task efficacy of control vectors for Mistral-7B-Instruct. This means that we apply control vectors trained on the bAbI task when evaluating on GSM8K and vice-versa.
>
> We have also added an additional section in the discussion (4.2) discussing how the specific tasks analyzed relate to the concept of reasoning. Both additional experiments indicate that the control vector intervention works which we believe strengthens our claims and the general results of the paper.
>
> **Weaknesses**
>
> 1. Contrastive pair selection
>
> The reviewer correctly highlights the various schemes and the issues with each of them. We tested schemes 2 and 3 by training control vectors based on them and observed that for both IOI and bABI the scheme 3 could improve performance more robustly. Interestingly, the additional experiments with GSM8K showed scheme 3 working better. In the updated manuscript we describe this in more detail for the additional experiment and also report results for scheme 2 with Mistral-7B-Instruct on GSM8K in the appendix to showcase the difference in contrastive pair schemes. We hope the reviewer finds this sufficient.
>
> 2. Qualitative errors
>
> A more qualitative analysis would indeed improve the paper. In the resubmitted manuscript we have added an appendix doing exactly this, but have focused on the additional experimentation done on GSM8K and Mistral-7B-Instruct. These experiments clearly show changes to the reasoning traces generated by the model that leads to correct reasoning when the intervention is applied. These results can be seen in A.3.
>
> **Questions**
>
> 1. Is reasoning task-specific?
>
> Good question. In the original manuscript (prior to the addition of GSM8K ) it would seem that reasoning is task-specific, but the new results suggest that the control vector might actually model a more general representation. In the updated manuscript we report that control vectors trained on bAbI improve model performance on GSM8K and vice-versa. We are hesitant to conclude that we have a found a general representation for reasoning, but it is a very interesting finding warranting more research into how reasoning ability is represented in the residual stream. See the updated results section for further discussion.
>
> 2. Which types of incorrect reasoning are reduced when using the method?
>
> By the reviewer’s suggestion we have tested the methodology on an OOD reasoning task. We have introduced GSM8K, as an example of mathematical reasoning. We have reported that control vectors trained on bAbI improve model performance on GSM8K and vice-versa.
>
> - *With regard to the minor points:*
>
> 1. Construction of control vector
>
> In our original submission, we attempted to explain the methodology in Section 2. In lines 126 - 128 we mention: "We extract the representation after each layer at the final token of a task example. From these extracted activations we derive layer-specific control vectors.".
>
> 2. Directionality of improvement
>
> This is correct, due to the PCA process the improvement is sometimes found by scaling alpha by a negative value. If one finds it important that a positive scale represents a “positively improved outcome”, then the direction can be rectified by flipping the direction after training the control vector based on empirical observation on a validation set.
>
> 3. Logit-based evaluation
>
> We can see that this is not fully clear. We further clarified our previous description slightly to make it clearer. So in lines 222 - 223 we state: “We benchmark performance by investigating the output logits of a constrained set of potential responses, where the potential responses are the collection of all answers in the dataset, similarly to what is done in the lm-eval framework.”.
>
> 4. KL divergence metric
>
> This is a correct observation and has been modified by the authors in the updated manuscript.
>
> We greatly appreciate the time and effort the reviewer (imCU) has spent on providing valuable feedback that has resulted in the submission of a stronger paper. We hope the reviewer agrees that the revisions and additional experiments we have performed have improved the paper and will lead the reviewer to reconsider their rating.

---

> > ### Comment · Reviewer_imCU · 2024-11-23
> >
> > I greatly appreciate the authors' additional experiments and thoughtful responses to my comments.
> >
> > As noted in the response (and in other reviewers' comments) I too am "hesitant to conclude that we have a found a general representation for reasoning,". The fact that CoT, for example, translates to better reasoning (at least in older models) is indicative of the fact that it is not completely unreasonable to expect that some ways of guiding the generation can elicit some performance gain, but I agree with reviewer TVGr that this is unlikely to be a "Superintelligence" hack", and understanding the limitations would be interesting. While I still have some reservations as to the actual statement that the control vector improves something that we can attribute to reasoning, these are outside the scope of this specific paper.
> >
> > I will be following the developing discussions here, but overall I support this paper for publication.

---

### Official Review · Reviewer_1E6m · 2024-10-29

**Soundness:** 3
**Presentation:** 3
**Contribution:** 3
**Rating:** 8
**Confidence:** 3

**Summary:**

The paper proposes a new way to improve reasoning in LLMs via representatoin engineering. It basically reads activations from the residual streams of LLMs during reasoning, derives a control vector from this, and applies this vector during inference time to modulate the representation space. The results are evaulated on inductive and deductive reasoning tasks using Pythia and Mistral Instruct 7B. This intervention seems to improve reasoning without further training, which suggests that reasoning is not too different from other information processing done by LLMs.

**Strengths:**

Originality - As far as I am aware, such control vector intervention is a new idea, and the proposed framework for deriving control vectors and their impact (KL div, entropy) is a useful contribution.

Clarity - The paper explains its methods and findings in a structured and thoughtful way. The transformer architecture for conceptually reframing this problem is helpful.

Significance - I think the result that reasoning is an emergent general information processing ability is useful and important.

**Weaknesses:**

1. The paper only tests the approach on two models. Larger models will need to be used to validate these findings.
2. I think the IOI task is too simple, even though the bABI task is representative. The authors should explore a wider range of reasoning tasks to strengthen the conclusions.
3. The contrastive approach using random strings seems a bit arbitrary. Probably a more principled method would be more fruitful.
4. The probability mass results seem to disagree a little with the entropy KL divergence results.

**Questions:**

1. How does performance intervention compare to finetuning? I believe that would be a fair baseline.
2. Are there any hypotheses as to why there is disagreement between the KL divergence and entropy results?
3. Is performance intervention sensitive to layer choice? How so?

---

> ### Author Response · Authors · 2024-11-21
>
> We want to thank the reviewer (1E6m) for the positive feedback emphasizing the importance of the research and the clarity of the presentation as well as providing well-founded critiques. We have taken measures to rectify the few mentioned weaknesses to strengthen the paper. The major change we’ve made to the paper are the following additions:
>
> - We run control vector experiments on GSM8K with Mistral-7B-Instruct.
>
> - We evaluate cross-task efficacy of control vectors for Mistral-7B-Instruct. This means that we apply control vectors trained on the bAbI task when evaluating on GSM8K and vice-versa.
>
> We have also added an additional section in the discussion (4.2) discussing how the specific tasks analyzed relate to the concept of reasoning. Both additional experiments indicate that the control vector intervention works which we believe strengthens our claims and the general results of the paper.
>
> **Weaknesses**
>
> 1. Larger models needed to validate findings
>
> We agree that scaling to larger models is an important direction. We focused on demonstrating our method's effectiveness on medium-sized models for this work, and current results with Pythia and Mistral-7B provide evidence for the method's potential, as these models are large enough to exhibit some level of reasoning capabilities while being computationally tractable for a thorough analysis.
>
> 2. Exploring a wider range of tasks
>
> This is a fair critique, and we have taken steps to include GSM8K as an additional dataset for evaluation. We have additionally tested the method for generalizability by evaluating control vectors across tasks, i.e. applying a GSM8K control vector to the bAbI task and vice-versa as mentioned above. We concretely show that applying the bAbI derived control vector improves model performance on GSM8K by approximately 7% and that applying GSM8K to bAbI increased model performance by approximately 3%.
>
> 3. Approach to generate contrastive pairs
>
> Our initial use of random strings provides a baseline contrast that, while simple, effectively demonstrates the method's core principles. Our new GSM8K experiments have revealed that using incorrect reasoning traces as contrast vectors yields even stronger results, supporting our theoretical framework. This finding suggests a natural progression toward more principled contrast selection in future work.
>
> 4. Evaluation metrics (probability mass, entropy, KL divergence)
>
> The two plots are not completely aligned because they assess slightly different things. The KL divergence results are looking at the entire logit distribution whereas the probability mass plots only look at a constrained set of logits related to the potential answer tokens as explained in equations 7 and 8. It is therefore hard to make a direct comparison between the KL divergence and probability mass plots.
>
> **Questions**
>
> 1. Comparison with fine-tuning
>
> This would be interesting to look into. Comparing to a fine-tuning would be very interesting, and altough our method is training-free compared to fine-tuning it would nonetheless be useful to note whether the increased upfront overhead of fine-tuning a model could payoff in the long term.
>
> 2. Disagreement between the KL divergence and entropy results?
>
> We have attempted to clarify the wording as to explain the different evaluation metrics. There does not necessarily need to be complete “agreement” between the entropy and KL divergence plot. This is because the entropy measure indicates that the model becomes more “confident” when generating its response. KL divergence provides insight into how the probability mass changes but not in which direction and to which tokens. The disagreement between the metrics is thus not truly a disagreement. See the second-to-last paragraph of section 3.2 for an explanation of this.
>
> 3. Is performance intervention sensitive to layer choice? How so?
>
> This is indeed an important consideration. One could imagine a simple experiment that varies the layers and the alphas to asses this. The choice to use the middle layer was based on prior literature, which found that applying the intervention to middle layers is sufficient to induce changes in style. It is conceivable that the choice of intervention layer could vary greatly depending on the task at hand and is an important question for future research.
>
> We appreciate the reviewer's (1E6m) thoughtful feedback, which has helped us strengthen the paper through additional experiments and clarifications. The new results, particularly the cross-task efficacy demonstration, reinforce our core contributions while addressing the raised concerns.

---

### Official Review · Reviewer_vEQT · 2024-11-03

**Soundness:** 3
**Presentation:** 3
**Contribution:** 2
**Rating:** 6
**Confidence:** 2

**Summary:**

The paper uses control vectors with a residual stream added to transformers, trained using contrastive prompt pairs, to improve reasoning ability in a couple of tasks. Some improvement is achieved with some ranges of a controlling parameter alpha.

**Strengths:**

- This is an insightful technique that can shed light on the behaviour of LLM and reasoning.
- Well written.

**Weaknesses:**

- Novelty. They apply an existing technique, adapting it to a couple of architectures and deriving prompts pairs for

- The results are only slightly better for some ranges of alpha.

- Very few models, very small models, and just two datasets.

- The insights about reasoning are somewhat limited. They say that "reasoning is not neccesarily different from any other form of processing done by LLMs". This is either a tautology if we are assuming that LLMs do reasoning or needs to be compared to some other kinds of specific processing that are not reasoning, such as perhaps retrieval, etc. If the meaning is that the patterns are no different, then the paper should compare with the other patterns. In any case, for two tasks, it's limited how much we can extrapolate for the understanding of what reasoning is.

- The second task (IOI) is too simple, and in some cases, as the authors say, the answer is not fully specified.

Minor
- "illustrated in 1" -> "illustrated in Figure 1"

**Questions:**

- How can we find the best alpha in a new deployment situation?

- How does it compare to chain-of-thought or a fine-tuning that highlights the prompts that are associated with good reasoning and bad reasoning? Can a similar information be provided contextually?

- Why do the contrastive pairs have to be associated with good and bad prompts instead of good and bad reasoning instance?

- What do you mean by: "We furthermore cannot expect to be able to improve a model on a task it
cannot correctly solve when the method of improving a model is derived directly from it’s [sic] own
hidden states."? The whole purpose is to improve the model on instance it wouldn't solve without the residual stream?

---

> ### Author Response · Authors · 2024-11-21
>
> Thank you (vEQT) for the feedback. We have substantially strengthened the paper through two major additions:
>
> We run control vector experiments on GSM8K with Mistral-7B-Instruct.
>
> We evaluate cross-task efficacy of control vectors for Mistral-7B-Instruct. This means that we apply control vectors trained on the bAbI task when evaluating on GSM8K and vice-versa.
>
> We have also added an additional section in the discussion (4.2) discussing how the specific tasks analyzed relate to the concept of reasoning. Both additional experiments indicate that the control vector intervention works which we believe strengthens our claims and the general results of the paper.
>
> **Weaknesses**
>
> 1. Novelty
>
> While the technique has correctly been used before, this work makes two novel contributions: (1) demonstrating that reasoning capabilities can be modulated through control vectors, similar to stylistic attributes, and now (2) showing cross-task generalization of these control vectors. By assessing “reasoning” in this manner we attempt to test whether reasoning performance can be modulated similarly to how the approach has been used to induce a certain generative style. There is a tendency to uplift reasoning as a key ability of any intelligent system and thereby demarcate reasoning from other types of information-processing tasks. The novelty is thus in testing whether reasoning can be modulated in a similar fashion to other standard information-processing tasks. To our knowledge, this specific research has not been attempted before. We furthermore believe that we have strengthened the claim that the research is directly related to reasoning performance with the addition of the GSM8K dataset.
>
> 2. Only slight increases in accuracy and few models
>
> The method achieves consistent improvements across multiple models and tasks, including larger relatively capable models like Mistral-7B-instruct. These gains are achieved without additional training or computational overhead, unlike alternative approaches. The results should also be seen in the context of general research in ML and NLP where one often sees new methods reporting increases in accuracy from training entire new architectures or models of a few percent. We additionally wouldn’t expect all \alpha to influence the models similarly as this is a hyperparameter that controls the magnitude of the control vector.
>
> As mentioned, we have additionally added further experiments showcasing that efficacy of the approach on the GSM8K dataset as well as experiments showcasing that the derived control vectors generalize across task to some degree. The specific results are discussed in the resubmitted paper in section 3.4 (lines 307 - 323) alongside updated plots and figure 6 showcasing qualitative examples of changes to reasoning traces for Mistral-7B-Instruct.
>
> 3. Insights about reasoning (reasoning being similar to other types of processing)
>
> We agree that IOI is a relatively simple task, although we would argue that even though they are simple they do necessitate some form of “reasoning”. We emphasize this point in an additional section of our discussion (section 4.2). We have as mentioned above furthermore added experiments on GSM8K with positive results, suggesting that the intervention works even for complex tasks as well as across tasks. The review highlights the claim that “reasoning is not necessarily different from any other form of processing done by LLMs” as potentially problematic and we have now rephrased this in the paper to state that certain aspects of reasoning seem to be encoded in the residual stream similarly to other tasks referenced in the paper (final sentence of abstract). We cannot conclude the previous statement, although the fact that we can modulate performance on reasoning benchmarks in a similar way is suggestive of this.
>
> The comparison we perform is essentially to previous work showcasing that the representation engineering works for inducing specific generative styles such as producing especially emotional text.

---

> ### Author Response · Authors · 2024-11-21
>
> **Questions**
>
> 1. How can we find the best alpha in a new deployment situation?
>
> Determining optimal values of \alpha would indeed be very interesting, and is an interesting avenue for future research. As it stands, the optimal values differ based on the model and the dataset at hand. This is not an issue in practice, since one can determine an optimal value empirically before use. The process would consist of finding the scale and direction of improvement on a development set before deployment. Nonetheless, this was not the main focus of this paper, although it could be interesting to analyze whether there exists any “laws” regarding optimal \alpha values based on the depth of the model, the size of the residual stream or perhaps the type of task.
>
> 2. Comparison with CoT and fine-tuning
>
> Comparing to fine-tuning would be very interesting, and although our method is training-free compared to fine-tuning. It would nonetheless be useful to note whether the increased upfront overhead of fine-tuning a model could pay off in the long term. It might also have made sense to compare our method to simply prompting a model to use CoT. CoT can, however, be token-heavy. Our method, in contrast, relies on no additional token usage.
>
> As an additional interesting direction for future research we have considered whether the technique could be used to induce CoT generation without CoT prompting.
>
> 3. Contrastive pairs
>
> As you correctly point out the contrastive pairs need to be based on a correct and incorrect example. We highlight a few different approaches to constructing contrastive pairs on lines 205 - 210.
>
> - Scheme 1: induce a model to perform bad reasoning
>
> - Scheme 2: use examples the model can solve vs. cannot solve
>
> - Scheme 3: use random strings
>
> With regards to scheme 1, the argument is that if you ask a model to do bad reasoning and it then correctly performs bad reasoning and incorrectly solves a task (from which we extract the hidden representation), the model as actually “correctly” answered the question wrong via a deliberate bad reasoning trace. For the additional experiments on GSM8K we test both scheme 2 and 3 and observe that scheme 2 is better at improving performance. There is also the fundamental difference in Mistral-7B-Instruct being an instruction fine-tuned model giving us the opportunity to actually assess the reasoning traces generated by the model.
>
> 4. Explanation of “Improving on task it cannot solve”
>
> This criticism is fair and is caused by an ambiguity in our wording. When we state that we “cannot improve a model on a task it cannot solve” it is because the control vectors are derived from the model’s own hidden dimension representation on task examples that it correctly solves. So if a model cannot solve a single example of a given task (or cannot do better than chance-level), we do not have any good hidden dimension representation examples from which we can derive the control vectors. The ability to push a model towards performing better on a task, relies on the notion that a model knows more about the task than its performance admits.
>
> The improvement we refer to is thus to use the representations extracted from the task examples that a model can correctly solve to induce “correct reasoning” on the examples from the task the model cannot correctly solve. Additionally we wish to emphasize that the residual stream is not a new addition, we merely add the intervention on the residual stream. The residual stream represents the hidden dimensional space of the model, allowing us to extract and modify reasoning patterns without architectural changes. The control vectors that we derive are based on the hidden dimension representation (read from the residual stream of a model). This is why our ability to improve a model on a task is dependent on a model's ability to initially solve some tasks of that form.
>
> We sincerely appreciate the time and effort the reviewer has dedicated to providing feedback on our paper. We hope the revisions adequately address the reviewer’s concerns and look forward to any further feedback you may have.

---

> > ### Comment · Reviewer_vEQT · 2024-11-24
> >
> > Thank you for the comments to my review and the other reviews as well. Very detailed, as well as the effort to improve the paper. I will raise my score slightly as the responses clarify the applicability of the method. I still have issues about the conceptualisation of "reasoning" in the whole paper, but that's a principled question that would require a more cognitive approach to the paper rather than an informal appreciation of benchmarks that have more or less "reasoning".

---

### Official Review · Reviewer_kzoH · 2024-11-04

**Soundness:** 2
**Presentation:** 3
**Contribution:** 2
**Rating:** 6
**Confidence:** 4

**Summary:**

This paper proposes improving reasoning performance of LLMs through representation engineering. The authors introduce the "control vector" derived from the residual stream of the model as an inference-time intervention, which modulates the representational space to enhance reasoning capabilities without additional training. The approach is evaluated on inductive (indirect-object identification) and deductive (bAbI) reasoning tasks with models from the Pythia suite and Mistral-7B-Instruct. The authors suggest that the intervention improves task performance and reasoning may not fundamentally differ from other token generation tasks implied by the results.

**Strengths:**

**Originality:** The representation engineering approach utilizing the residual stream for reasoning task offers a novel view.

**Quality:** The study is methodologically clear. The use of models at varying scales and the combination of several metrics (accuracy, KL divergence, entropy) offers a comprehensive evaluation.

**Clarity:** The paper clearly explains the steps for deriving control vectors.

**Significance:** The problem of whether reasoning requires distinct treatment than other tasks is important.

**Weaknesses:**

1. **Task complexity**: The reasoning tasks (IOI and bAbI) used are relatively simple, potentially limiting the conclusions drawn about "reasoning" in general. Results might differ on more challenging tasks. Specifically, mathematical reasoning has received great attention as a difficult type of reasoning and has been widely adopted as benchmarks for LLMs. Could you add addtional experiments on the GSM8K or MATH dataset to explore how this approach applies to more complex reasoning tasks?
2. **Model scope**: Testing is limited to smaller models (up to 7B parameters), which could restrict applicability of the conclusions. Larger models might capture reasoning dynamics differently and yield varied results under similar interventions. Could you include experiments for Pythia-12B to investigate how the method scales with model size?
3. **Hyperparameter choice:** As indicated by the experimental results, the choice of $\alpha$ actually dominates the final outcome. However the paper does not provide a systematic procedure to choose $\alpha$ beforehand. To make the framework applicable to a wide range of tasks and models, could you provide a general routine to determine the value of $\alpha$ and its interpretation?
4. **Contrastive pair selection and control vector derivation:** The paper uses random character strings as "negative" samples and PCA-based control vectors according to empirical performance. I wonder if we can gain more insights for these choices. Could this empirical advantage be due to the nature of IOI task, where answers could be semantically correct but not expected? This is also why I suggest to include math reasoning in point 1, where ambiguity of answers is minimized. Could you report the results for different types of contrastive prompts (e.g., partially correct responses) on the GSM8K / MATH dataset to investigate how they impact control vector efficacy?
5. **Experiment settings:** The Pythia models are only reported for the IOI task, which seems insufficient to me. Could you provide results of Pythia-1.4B & Pythia-2.8B on the bAbI task also?
6. **Result interpretation:** If I understand the figures correctly, there is an apparent decrease for B & BL condition and almost no improvement for A & AL condition with the Pythia-2.8B model in logit-based accuracy. This raises my doubts about the effectiveness of the approach and is probably my biggest concern. The performance gap between A condition and B condition may imply the statistical learning bias in disguise rather than genuine reasoning. Therefore the conclusion that "there is no intrinsic difference between the process of reasoning and other information-processing tasks performed by LLMs" does not seem valid to me. Could you make further explanation regarding this experiment result?

**Questions:**

See Weaknesses above.

---

> ### Author Response · Authors · 2024-11-21
>
> We appreciate the reviewer's (kzoH) thorough feedback highlighting both the strengths and limitations of our work. We have made substantial improvements to address these concerns, particularly through additional experiments that strengthen our core claims. The major changes we have made to the paper are the following additions:
>
> - We run control vector experiments on GSM8K with Mistral-7B-Instruct.
> - We evaluate cross-task efficacy of control vectors for Mistral-7B-Instruct. This means that we apply control vectors trained on the bAbI task when evaluating on GSM8K and vice-versa.
>
> We have also added an additional section in the discussion (4.2) discussing how the specific tasks analyzed relate to the concept of reasoning. Both additional experiments indicate that the control vector intervention works which we believe strengthens our claims and the general results of the paper.
>
> 1. Task Complexity
>
> We agree with this critique, although we would argue that even though they are simple they do necessitate some form of “reasoning”. We emphasize this point in an additional section of our discussion (section 4.2). As you suggested, we added further experiments on GSM8K, showing positive results and suggesting that the intervention works even for complex tasks as well as across tasks. See also the addition of figure 6 top of page 9 showcasing how reasoning traces are affected for Mistral-7B-Instruct on GSM8K.
>
> 2. Model Scope
>
> The models are somewhat limited, especially the Pythia-suite. Models such as Mistral-7B outperform larger models such as pythia-12b that are trained on the exact same amount of data in the exact same order as the smaller Pythia models. We thus thought it would be more interesting to do additional research on the more complex tasks you suggested, namely GSM8K, for the larger model analyzed in the paper.
>
> 3. Hyperparameter Choice
>
> This is indeed an important aspect to be careful and systematic about. As it stands, the optimal values differ based on the model and the dataset at hand. This is not an issue in practice, since one can determine an optimal value empirically before use. The process would consist of finding the optimal \alpha and direction of improvement on a development set before deployment. More generally you would simply scale alpha until no further improvements can be made on the development set.
>
> This was, however, not the main focus of this paper, although it could be interesting to analyze whether there exists any “laws” regarding optimal \alpha values based on the depth of the model, the size of the residual stream or perhaps the type of task.
>
> 4. Contrastive Pair and Control Vector Derivation
>
> You raise an important point related to the derivation of the control vectors based on contrastive pairs. In the paper (line 207 - 210) we discuss the issue of using good / bad reasoning examples as the contrastive pairs and thus end up using the random strings as the negative set of the contrastive pair with some success. It could, as you highlight, be problematic for the IOI task where incorrect answers aren’t necessarily “wrong”, although this shouldn’t be an issue with bAbI or GSM8K. There is furthermore the limitation with both IOI and bAbI that the responses are essentially given as single words and not reasoning traces. This is addressed with the addition of GSM8K, where we apply both scheme 2 and 3 as discussed in the paper (correct vs. incorrect reasoning and random strings). We report the performance of both schemes in the updated paper and showcase the for Mistral-7B on GSM8K scheme 2 actually works better in terms of improving performance.
>
> A more qualitative analysis of the reasoning traces generated by Mistral-7B-Instruct would be very interesting but was unfortunately out of scope for the paper. We have added a few qualitative examples of reasoning traces (see figure 6 top of page 9 in the updated manuscript) to address this comment.

---

> ### Author Response · Authors · 2024-11-21
>
> 5. Experiment Settings
>
> The limitation to IOI tasks for Pythia models reflects a fundamental capability constraint: these models perform at near-chance levels on bAbI tasks, making them unsuitable for meaningful analysis of the intervention's effects on task performance. This observation further motivated our focus on more capable models for the additional experiments. We have added an additional note on this in the results section of the updated paper (line 301 - 303).
>
> 6. Result Interpretation
>
> While the accuracy increases are modest, they demonstrate a consistent pattern of improvement across conditions when applying the control vector. Importantly, the differential effects across conditions (A, AL, B, and BL) provide insight into how the intervention affects the model's capabilities. In Figure 2a we observe that all conditions increase when we add the control vector derived from the A condition scaled by an \alpha of 0.2. The key observation here is that all conditions improve, albeit slightly. In Figure 3a we see that the A and AL conditions aren’t very strongly affected by the control vector although the control vector is derived from the A example prompts. We do however see a relatively strong influence on the B condition and the BL condition. The plots even suggest that we could have scaled the control vector with an \alpha higher than |1| to improve the performance even further, although this wasn’t done for the Pythia models. Another note on this is that accuracy improves for B and BL as we scale the control vector negatively. This might seem counter-intuitive, but is just an artifact of the control vector being derived from the PCA on hidden states and we thus lose the original sense of directionality that we had in the original positive / negative prompt pairs. This can either be rectified based on an evaluation after training, or can simply be considered part of the \alpha-selection process.
>
> Rather than focusing on further Pythia experiments, we ran the described additional experiments on GSM8K and cross-task control vector application. We discuss these results in-depth in the resubmitted paper (lines 307 - 323), but as stated we do improve performance on GSM8K and also see that a control vector trained on bAbI is effective for improving performance on GSM8K and vice-versa. See figures 4 and 5 in the updated manuscript.
>
> We sincerely appreciate the time and effort the reviewer (kzoH) has dedicated to providing thorough and thoughtful feedback on our paper. The suggestion to include GSM8K in our analysis has significantly strengthened both our results and the overall quality of the paper. We hope the revisions adequately address the reviewer’s concerns and look forward to any further feedback you may have.

---

> > ### Comment · Reviewer_kzoH · 2024-11-23
> >
> > I appreciate the authors' efforts to include additional experiments on GSM8K and cross-task efficacy, it helps strengthen the task complexity covered by this paper and fits better to the reasoning topic. However, point 3, 6 are not convincingly addressed. It seems to me that clearly in Figure 2a with $\alpha = 0.2$, the accuracy on condition B and BL drops, and the improvements on other conditions are negligible. The fluctuations on GSM8K further strengthen my doubts about whether we can find a meaningful value of $\alpha$ systematically for substantial improvement of reasoning capabilities. My concern that the performance change may imply a statistical learning bias in disguise rather than genuine reasoning remains.

---

> > > ### Author Response · Authors · 2024-11-26
> > >
> > > Dear Reviewer (kzoH),
> > >
> > > Thank you again for the feedback and for highlighting your concerns. We would like to clarify a few key points regarding the interpretation of the results. While you correctly note the decrease in accuracy for B and BL conditions at α = 0.2, we have found that applying a negative scaling (α = -1) produces improvements, concretely we observe:
> > >
> > > - $\sim 0.39 \rightarrow \sim 0.43$, for the *B* condition, and
> > > - $\sim 0.34 \rightarrow \sim 0.38$, for the *BL* condition.
> > >
> > > This is visible in the *Figure 2a* as B and BL decreasing in accuracy for positive alphas but increasing for negative alphas. During control vector creation the sign of the alpha could theoretically be changed, if one desires, so that positive alphas always mean positive influence on the accuracy. We will attempt to implement this functionality in the library before publication in order to correctly convey the bidirectionally/symmetry of the control vector activations, and to reduce potential misunderstandings of the graphs results (i.e. avoiding misinterpretation that control vectors perform poorly on the B & BL task; they don’t, they just show improvement in the negative direction). We furthermore observe similar improvement on the GSM8K questions analyzed.
> > >
> > > The bi-directionality of the control vectors further suggests to us that the improvements seen across conditions are not entirely due to statistical learning biases. Since the control vectors are derived from applying PCA to the differences in representations between the contrastive pairs, the resulting sign of α is arbitrary - what matters is the magnitude and direction of the induced changes. We agree that we cannot conclude whether this has to do with genuine reasoning. In addition to section 4.2 (which we've added in the revised paper) wherein we discuss the reasoning as a concept in relation to our research, we have also attempted to tone down claims about reasoning throughout the paper. We aim to address these concerns in future work by focusing on more capable models as well as by expanding the approach to more varied tasks. We furthermore wish to answer why control vectors of certain magnitudes work better at modulating the ability to answer correctly on these specific tasks than others. It could very well be that modulating the representation via this intervention is in part possible due to statistical biases in the training data (as also highlighted by reviewer TVGr), but even if that is the case, we believe it interesting and important that this can for certain questions modulate a model's response. Rather than a limitation, this highlights the utility of representation engineering as a principled way to guide the model toward desirable outputs without retraining. The preliminary results we show indicate that the actual "reasoning" done by the model (assessed from a purely behavioristic aspect via the reasoning traces provided by the model on GSM8K) does change when the intervention is applied. We find the qualitative examples added (figure 6 top of page 9) illustrative. We have additionally added an example in the appendix (Figure 8) of an incorrect example.
> > >
> > > Again, thank you for the critique and in-depth feedback you have provided. It has helped in strengthening the paper as well as posing interesting questions for our future work.

---

> > > > ### Comment · Reviewer_kzoH · 2024-11-26
> > > >
> > > > I really want to thank the author for the efforts. Regarding the IOI task, it is not the **negative** value of $\alpha$ for B and BL that concerns me, but rather we cannot find an $\alpha$ which **simultaneously** improves A and B. For $\alpha = -1$, the performance on B increases, but the performance on A drops, and we are applying the **same** control vector derived from A condition. **Do we need to find a different value of $\alpha$ for each different task in this case?** I am happy to raise my score if this is a misunderstanding.

---

> > > > > ### Author Response · Authors · 2024-11-26
> > > > >
> > > > > Thank you for the clarification! You are correct that accuracy drops slightly for the A condition while performance improves for B and BL and remains steady for the AL condition which is not really affected.
> > > > >
> > > > > As you (and others) correctly pointed out in the original review the IOI task is very simple, and the results observed in Figure 2 are likely caused by this simplicity. The IOI task merely involves finding the correct indirect object, and the change in conditions between A (and AL) and B (and BL) is the location of the indirect object. What this could mean is that the IOI control vectors (due to the simplicity of the task) are encoding the ability to find a specific entity (name) at a specific location in the context (corresponding to the B and BL condition). This position-sensitive encoding might explain why we see different optimal $\alpha$ values for different conditions. It also hints quite clearly at control vectors being sensitive to the complexity and variety of tasks they are trained on. This aspect of scaling complexity of the tasks is what we partly discuss in section 4.2.
> > > > >
> > > > > Your question as to whether we need to find a different value for each task in IOI is important and what we attempted to address with the cross-task experiments on more complex tasks. Applying the approach to tasks such as bAbI and now GSM8K provides us with a more varied and potentially rich hidden state from which the control vector can be derived, which we hypothesize contains a more representative control vector than the IOI task. This is also what we believe the cross-task experiment hints at; using more complex tasks results in encoding more interesting and rich information with regard to the ability to solve tasks like bAbI and GSM8K.
> > > > >
> > > > > We have modified the manuscript slightly in the results and limitations to make note and speculate about of the observation concerning the lack of cross-task generalization on the IOI task.
> > > > >
> > > > > We have made the following modifications:
> > > > >
> > > > > - **Addition to results**
> > > > >
> > > > > "Pythia-2.8B improves accuracy slightly when the control vector is applied across conditions, with some variation in $\alpha$, and KL divergence increases quadratically as $\alpha$ is increased in both directions, see figure Figure 2a. These findings are corroborated by (...)"
> > > > > -->
> > > > > "Pythia-2.8B improves accuracy slightly when the control vector is applied across conditions, with some variation in $\alpha$, and KL divergence increases quadratically as $\alpha$ is increased in both directions, see Figure 2. The intervention is mostly effective for B and BL suggesting that the control vectors are encoding position-sensitive information of the indirect-object to be generated. These findings are corroborated by"
> > > > >
> > > > > - **Addition to limitations**
> > > > >
> > > > > "The IOI reasoning task is furthermore a relatively simple task, which has both advantages and disadvantages. The task is mostly interesting to our research when models cannot correctly answer every question, a very hard balance to strike."
> > > > > -->
> > > > > "The IOI reasoning task is furthermore a relatively simple task, which has both advantages and disadvantages. The results indicate that a task such as IOI may even be *too* simple to elicit interesting representations capturing a more general phenomenon. The task is mostly interesting to our research when models cannot correctly answer every question, a very hard balance to strike."Still, these results point to many interesting research questions that we are excited to pursue; some of them related to the robustness of the $\alpha$ parameters and potential guidelines for how to optimally choose $\alpha$ based on model size, task complexity and other dimensions.

---

> > > > > > ### Comment · Reviewer_kzoH · 2024-11-26
> > > > > >
> > > > > > Thank you for the response! If I understand correctly, in the cross task experiment, the control vector derived from bAbI needs $\alpha = 2$ to achieve performance gain on the bAbI task (Figure 3(a)), but $\alpha=-1$ on the GSM8K (Figure 5(a)). This indicates that we need to choose different $\alpha$ values (if not control vector) for each different task. However for a model with *improved reasoning performance*, I would expect a consistent improvement across different tasks, instead of gaining better performance on one task, but resulting in worse performance on another task, like a seesaw. I wonder if you can plot a figure where a derived control vector with $\alpha$ values can achieve performance gain **simultaneously on different tasks**, such as bAbI and GSM8K. If this concern can be addressed, I would happily vote for strong accept.

---

> > > > > > > ### Author Response · Authors · 2024-11-27
> > > > > > >
> > > > > > > It is unfortunately not feasible to process and visualize the suggestion in time. But we wish to emphasize that it is expected that a control vector should be applied with different $\alpha$ values because the model representations (which are what we intervene on) have different starting points in the representational space dependent on the task.
> > > > > > >
> > > > > > > A potential experiment to illustrate this phenomenon could be designed as follows: Consider taking the representations from bAbI and GSM8K examples and calculating the centroid (mean representation) for each cluster. By applying a control vector scaled negatively to one centroid and positively to the other, we could then measure the distance between the centroids under two conditions: (1) before the intervention and (2) after the intervention. If we observe that the distance between the bAbI and GSM8K centroids decreases following the intervention, it suggests that these groups initially occupied distinct regions in the space and that the 'reasoning' vector needs to push the centroids from different starting points towards a "good at reasoning" point.
> > > > > > >
> > > > > > > * Take the following toy example:
> > > > > > >
> > > > > > > $X_{babi} = [1, 1, 2]$, the model representation at a given layer for a bAbI example
> > > > > > >
> > > > > > > $X_{GSM} = [1, 1, -3]$, the model representation at a given layer for a GSM8K example
> > > > > > >
> > > > > > > $cv = [0, 0, 1]$, the control vector trained on either of the tasks
> > > > > > >
> > > > > > > * For $\alpha = 2$
> > > > > > >
> > > > > > > $X_{babi} + \alpha \cdot cv = [1, 1, 4]$
> > > > > > >
> > > > > > > $X_{GSM} + \alpha \cdot cv = [1, 1, -1]$
> > > > > > >
> > > > > > > * For $\alpha = -2$
> > > > > > >
> > > > > > > $X_{babi} + \alpha \cdot cv = [1, 1, 0]$
> > > > > > >
> > > > > > > $X_{GSM} + \alpha \cdot cv = [1, 1, -5]$
> > > > > > >
> > > > > > > Suppose the theoretically optimal representation for a model to improve its performance on a given task is $\sim [1, 1, 1]$. Then applying the $cv$ with $\alpha = 2$ would push the GSM representation closer to optimal while pushing bAbI representations further away, whereas applying the $cv$ with $\alpha = -2$ would result in the opposite. The important aspect here is that performance can seemingly be improved along the same axis in the representational space the model navigates as information is processed.
> > > > > > >
> > > > > > > With this example we attempt to explain our intuition concerning how the representations behave when the intervention is applied: the control vectors effectively adjust the model representation in distinct directions within the representational space, guiding them toward a common alignment. Note, this alignment occurs along one of infinite directions that the control vector could assume in a high-dimensional (4096-dimensional) space.
> > > > > > >
> > > > > > > Thank you again for making a good point. Deriving a control vector that modulated performance of any and all tasks along the same direction and magnitude would be a great finding. Nevertheless, being able to find an axis that modulates performance (even if this does entail figuring out what the appropriate value of $\alpha$ is for a given task) is interesting as well. Models in deployment regularly encounter domain shift, requiring retraining. In a deployment scenario one could imagine a similar scenario in which the performance was regularly monitored at different $\alpha$ scales, and changed appropriately as the domain / tasks shift.

---

> > > > > > > > ### Comment · Reviewer_kzoH · 2024-11-27
> > > > > > > >
> > > > > > > > I really want to thank the reviewers for diligently addressing my concerns. Actually I admit that deriving a control vector which modulates performance over different tasks would go beyond the scope of this paper. I think the finding is interesting overall, and the discussion process has been very instructive. I have raised my score accordingly.

---

### Meta-Review · Area_Chair_MRRj · 2024-12-11

**Metareview:**

This paper introduces a method to improve Large Language Model (LLM) reasoning by using a "control vector" derived from internal activations to adjust the representation space during inference. The problem is of significance and the approach is novel, experiments are diverse and well supported, paper is clearly written.

**Additional Comments On Reviewer Discussion:**

Most reviewers engaged well.

---

### Decision · Program_Chairs · 2025-01-22

Accept (Poster)